# Constitution of a multicentennial multirisk database in a mountainous environment from composite sources: the example of the Vallouise-Pelvoux municipality (Ecrins, France)

Louise Dallons Thanneur[1], Florie Giacona[1-2], Nicolas Eckert[1], Philippe Frey[1]

[1]Univ. Grenoble Alpes, INRAE, CNRS, IRD, Grenoble INP, IGE, 38000 Grenoble,
[2]Laboratoire de recherche historique Rhône-Alpes (LARHRA) – UMR 5190

*Correspondence to*: Florie Giacona (florie.giacona@inrae.fr)

**Abstract** To develop efficient mountain risk management strategies, an obvious, yet tremendously difficult prerequisite is the constitution of comprehensive databases of past events and their impacts over long-time frames. However, existing records are often too short and siloed between different data providers and/or as function of hazards. To fill this gap, a methodology based on the combination of scattered pre-existing records with further archival research is proposed and used to populate a well-structured multirisk database covering the period 1600-2020 AD in a municipality of the French Alps – Vallouise-Pelvoux. Results include 2131 events related to rockfall, landslides, snow avalanches, floods (including debris flows) and glacial hazards, with documentation of possible interactions between hazards, their characteristics and detailed impacts. The combined use of different sources - and in particular archival searches - and their cross-referencing therefore provides a detailed record of past events that goes far beyond any inventory existing at the local scale. The analysis suggests that the distribution of events results from the combined effect of hazards, sources and human activities putting assets at risk, with a primary effect of sources. The methodology opens perspective for multirisk assessment in mountain territories and can be usefully transferred to other case studies.

## 1 Introduction

Mountain environments are subject to highly damaging hazards, both hydrological (e.g., floods and debris flows) and gravitational (e.g., snow avalanches, hereafter denoted avalanches, landslides, ice and rockfall). Interactions with societies and their practices put various assets at risk: settlements and their inhabitants, mountain practitioners, critical infrastructures, forest and ecosystems, etc. (Bründl et al., 2009; Eckert et al., 2012). Also, climate change is having a significant impact on mountain environments (Beniston et al., 2018; Pepin et al., 2022), with more extreme precipitation, higher temperatures, glacier retreat and permafrost thawing that modify magnitude, frequency and terrain prone to hazards (Hock et al. 2019; Jacquemart et al., 2024). In parallel, ongoing social changes including changes in population and social practices are rapid and strong (Altaweel et al., 2015; Lavorel et al., 2023). These combined physical and social dynamics lead to the emergence of new risk situations, which notably threaten densely populated mountainous valleys of the European Alps (Hock et al. 2019; Huggel et al., 2019; Eckert and Giacona, 2023). A specific concern is related to complex and/or cascading hazards that can have far-reaching and multiple consequences downstream (e.g., Vincent et al., 2010; Allen et al., 2016), for example floods hitting downstream settlements and infrastructures after glacier lake outburst with entrainment of alluvial material from the glacier margins. To achieve sustainable management of mountain territories, it is therefore essential to consider and understand the spatio-temporal variability of mountain hazards and related risks on the long range (Fuchs et al., 2013; Zgheib et al., 2022), including potentially complex dependencies in space, time and within hazards.

In the field of natural hazards, risk is classically defined in the IPCC model as a 'trefoil' marking the intersection of hazard, vulnerability, and exposure components (Reisinger et al., 2020). Recently, a multitude of terms has been proposed to account for increasingly complex hazards and risks, e.g. multihazards, cascading hazards, domino effects, compound events, cascading

risks, interconnected risks, etc. (e.g. Simpson et al., 2021; Zscheischler et al., 2018). Despite the slight difference in meaning
between these different terms, they all emphasize the importance of interactions between hazards and/or exposed elements.
Here, as in Giacona et al. (2017a; 2019b), risks related to natural hazards are defined as the results of interactions between
natural and societal components, each of these being characterized by their own temporal dynamics. This definition can be
seen as an expansion of the classical risk conceptualisation from IPCC, with explicit emphasis on interactions and temporal
dynamics, allowing a multirisk approach (Curt, 2021) to be adopted in a straightforward way. We use the multirisk concept in
a broad sense, namely all single risks and their interactions at the territorial scale, with interactions concerning potentially
hazards and/or impacts. Similarly, following UNDRR (2017) we define multihazards as all single hazards and their interactions
at the territorial scale.

In this context, an obvious, yet tremendously difficult prerequisite to efficient mountain risk management strategies is the
constitution of comprehensive databases of past events over long-time frames as exhaustive as possible (e.g., Jacquemart et
al., 2024). Here, 'event' means the occurrence of a given hazard and its characteristics, including impacts retrieved from
sources (Giacona et al., 2022). In many mountain areas, data for the most recent periods are indeed numerous, since they are
now often systematically collected and preserved digitally, but such data generally do not exist before the twentieth century
(e.g., Eckert et al., 2024). Also, existing series often result from partial inventories by different services or institutions
according to their specificities (e.g., road or forest management) and/ or at a large scale. As a consequence, none of these
existing inventories is exhaustive for a given hazard at the territorial scale, and they cannot be used to assess the evolution of
hazards or risks without an in-depth critical analysis (Giacona et al., 2019a; 2022; Maanan et al., 2022; Athimon et al., 2022).
Finally, most of existing (rather) long series of events related to mountain risks focus on a single hazard, (e.g., Stoffel et al.,
2005; Wilhelm et al., 2022; Boisson et al., 2022), with the few exceptions that cover especially short time periods (Martin,
1996). This makes it impossible to account for dependencies between hazards and risks, thus precluding for a comprehensive
risk assessment (Pescaroli and Alexander, 2018). Hence, producing records spanning several centuries and sufficiently
comprehensive to capture complex dynamics still requires considerable efforts to combine and standardise different data
sources, and getting rid of silos between hazards and data producers and their specificities.

More specifically, a commonly used strategy to reconstruct long series of past events is to use proxy-sources such as tree-rings
or sediment lakes (Stoffel and Bollschweiler, 2008; Schläppy et al., 2013; Wilhelm et al., 2022). However, resulting time series
are generally extremely local (e.g., avalanche activity in a single path, or rockfall activity on a single talus slope), and no
information on damage and, hence, risk is available (Favillier et al., 2018; Mainieri et al., 2020). The classical alternative to
proxy sources is historical sources (Wilhelm et al., 2019; Giacona et al., 2019b). Event chronologies based on historical sources
can be used to analyse spatial and temporal changes in potentially damaging phenomena: on the one hand, on their probability
of occurrence (i.e., the hazard, Zhong et al., 2024); on the other hand, on the associated damage (i.e., the risk, Garcia Hernandez
et al., 2017). Such chronologies have also assessed the impact of climate change on hazards, and, therefore, risks (Graham et
al., 1997; Flageollet et al., 1999, Barriendos et al. 2019; Duquesnes and Carozza, 2019; Giacona et al., 2021). However,
chronologies of events from historical sources have their own biases with regards to the reality of risks and hazards (Laternser
and Schneebeli, 2002; Giacona et al., 2019b). In the frequent case where partial databases and records already exist, it is
necessary to identify i) how inputs from archival searches and these sources can be cross-checked and combined, and ii) to
which extent the resulting composite chronologies of past events will be informative enough to analyse past changes in risks
and hazards at the territorial scale.

To address these issues, this paper develops a methodology to create a well-structured multirisk database as exhaustive as
possible on a multi-centennial time scale from composite sources that are usually not assembled; on the one hand, existing
databases including scientific, non-scientific and statutory documents mostly produced by public services and organizations
such as the French Forest Office (ONF) and its specialized Mountain Land Restoration service (RTM); on the other hand,
information resulting from intensive research in historical archives produced by different stakeholders and institutions. Our

database results from the combination and cross-checking of the different sources, providing extensive information on the location of each event, the characteristics of the hazards and a detailed description of the damage caused. As a proof of concept, the approach is applied to populate a multirisk event database over 420 years (1600-2020 AD) in the municipality of Vallouise-Pelvoux, a high Alpine valley in the Écrins massif, France. This territory was chosen because it is subject to various hazards occurring over a wide range of elevations. Notably it is marked by glacial recession, which generates complex cascading hazards, a peculiar case of multihazard where the glacial hazard triggers another hazard, often a flood. The database covers avalanches, the diverse types of floods occurring in the area (including debris flows), rockfall, landslides, glacial hazards and multihazards, namely various combinations of these hazards. The period from 1600 to 2020 includes various climatic conditions: the Maunder minimum (c. 1650-1700) which is the coldest period of the last millennium, the end of the Little Ice Age (1850-1900), the natural Early Twentieth Century Warming (c. 1900-1940) and the recent accelerated anthropogenic warming. This diversity makes the study period highly insightful regarding hazard-climate relations. In parallel, the valley has been inhabited over the whole study period, with a diversity of activities and a rich history. This has resulted in numerous and changing assets at risk, generating archival sources by various stakeholders potentially informative regarding natural hazards. The Vallouise-Pelvoux territory is therefore suitable to demonstrate the potential of the approach for providing a record of past events that, in a mountain context, goes beyond any pre-existing inventory in terms of events number, diversity of risks, temporal coverage and detailed description. In the following, after presenting the study area (Sect.2), the method used to produce the Vallouise-Pelvoux multirisk database is described (Sect. 3). The dataset and a statistical overview of sources, events and their characteristics are then presented (Sect. 4). Finally, the main characteristics of the spatio-temporal distribution of the events and their drivers (Sect. 5) and potential outlooks (Sect. 6) are discussed.

## 2 Study area

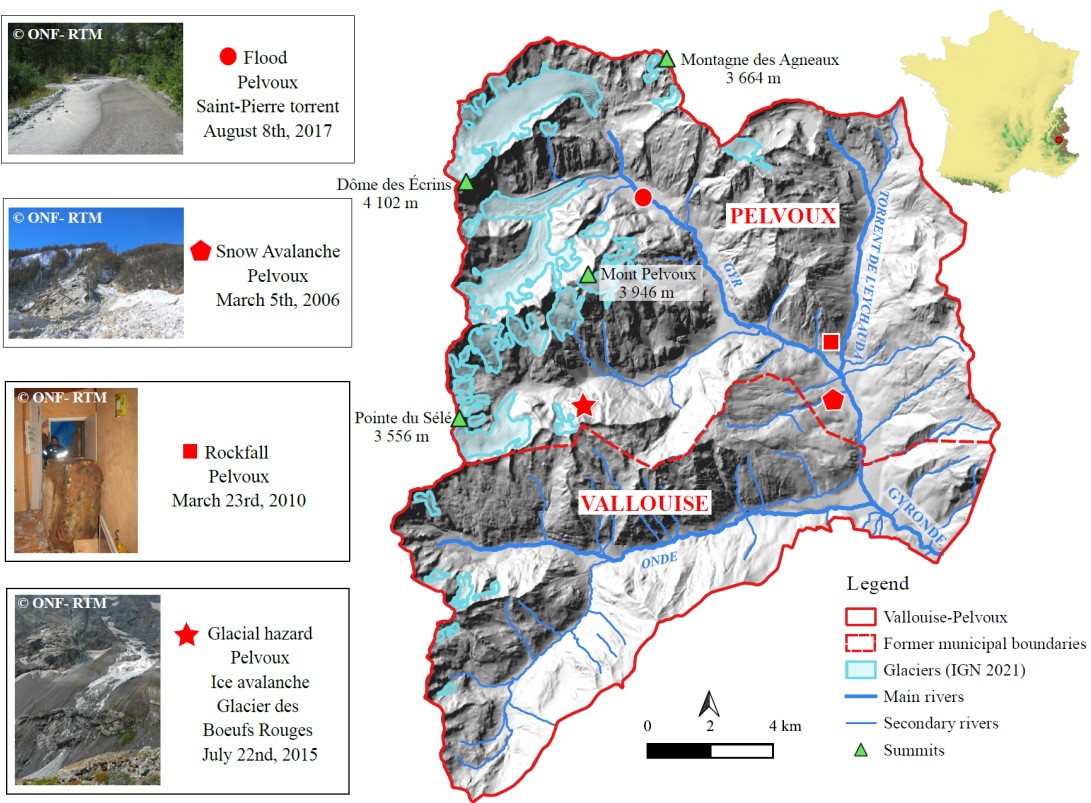

**Figure 1: Map and topography of Vallouise-Pelvoux Municipality with the former administrative boundaries of Vallouise and Pelvoux, main hydrological network, summits and glacier extensions in 2021. Left, examples of the main hazards occurring in the municipality: flood, rockfall (with example of impact on a dwelling house), snow avalanche and glacial hazard (ice avalanche). Pictures from ONF-RTM. DEM from IGN.**

## 2.1 Physical features

The municipality of Vallouise-Pelvoux is located in a high alpine valley in the southern French Alps (Figure 1). It presents a wide range of elevations, with a minimum of 1106 m, and a maximum of 4102 m a.s.l (Barres des Écrins) and a surface area of 145 km². Colas (2000) describes the climate of Vallouise-Pelvoux as mountainous with Mediterranean influences, resulting in dry summers followed by rainy autumns and winters during which snowfall are frequent, even at the lowest elevations. According to the Pelvoux weather station located at 1270 m altitude (latitude 44°52'14" N, longitude 6°29'13" E) over the period 1991-2020, the municipality had an average annual temperature of 8.3°C and an average cumulative rainfall of 929 mm (Météo France, 2024). The topography has a strong impact on the spatial distribution of precipitation, with often intense and localized rainfall. The municipality has a high insolation level and significant seasonal contrasts in terms of temperature and precipitations types. This harsh, contrasted, climate stimulates erosion processes, which, combined to the diversity of rocks and land cover, results in a wide variety of geomorphological forms (Colas, 2000).

In terms of geology, Vallouise-Pelvoux lies at the boundary between the inner Alps and the crystalline axial zone of the outer Alps (Debelmas, 2011; Colas, 2000). It is made of two superimposed mountain ranges of different geological origins. The first and oldest is the Hercynian chain that forms the relief to the west of the valley: the eastern slope of the Pelvoux massif. Built during the Primary Era, it is mainly constituted of crystalline rocks such as granites and gneisses. At the higher elevations, deglaciation is not yet complete because the hard rock has enabled glacial landscapes to be maintained. The second geological area, more recent, is part of the Alpine chain that forms the eastern side of Vallouise. It is composed of superimposed thrust sheets whose irregular deformation is responsible for the variety of outcrops. It is mainly constituted of limestone, but almost every type of rock is represented (Colas, 2000; Termier, 1903).

Vallouise-Pelvoux has a significant cryosphere (glaciers, permafrost and seasonal snow cover) with three main glaciers: the Glacier Noir, the Glacier Blanc and the Glacier du Sélé. Existing aerial photograph and satellite image processing (Defernand, 2021) resulted in a glacier area of 10.1 km² in 2018, namely 7 % of the total area. Following large scale trends (e.g., Zemp et al., 2019), these glaciers as well as the permafrost, are subject to the effects of climate change and are experiencing accelerating retreat (Veyret, 1952; Lardeux et al., 2016). Regarding snowfall and snow on the ground, long records do not exist locally, as in many areas. Yet, studies grounding on station data or reanalyses from the wider (southern) French Alps have demonstrated significant ongoing decreases over the recent decades due to warming (Durand et al., 2009; Verfaillie et al., 2018; Beaumet et al., 2021). However, at the highest elevations of the municipality, extreme snowfalls are still increasing because temperature are cold enough for precipitation to fall as snow (Le Roux et al., 2021).

The municipality is drained by three main rivers (Gyr, Onde and Gyronde) receiving many torrents which drain the slopes. By torrents, we mean streams of small catchment area, with a slope generally greater than 6%, with specific flood regimes often highly concentrated in sediment. The hydrological regime of all rivers and torrents is strongly influenced by glaciers and seasonal snow covers.

These physical characteristics generate a variety of hazards (Figure 1). The steep slopes of bare rock combined with frost action and intense precipitation are prone to rockfall and snow avalanches. Softer rocks and slopes filled by colluvium partially covered by vegetation imply landslides and a significant torrential activity including very concentrated, poorly sorted bedload transport and debris flows (Frey and Church 2009; Recking et al., 2013; Dudill et al. 2018; Jomelli et al., 2019). Changes in climate and the environment may have strong consequences on hazards. For example, glacial retreat and permafrost thawing favours land destabilisation and occurrence of glacial and periglacial hazards as well as multi hazards resulting from complex cascading processes. Different studies highlighted significant patterns of change in hazards in the surrounding French Alpine space (Eckert et al., 2013 for snow avalanches; Bard et al., 2015 for floods; Mainieri et al., 2023 for rockfalls), but none of them especially targeted the Vallouise-Pelvoux Municipality.

## 2.2 Socio-economics features

Vallouise and Pelvoux were two separate municipalities till 2017. Vallouise-Pelvoux has undergone many changes, both in terms of administrative boundaries, political regimes, land use and economy. Indeed, Vallouise-Pelvoux has been part of the historical and cultural region of Dauphiné since the 11th century, and along with other mountain territories, was part of the Grand Escarton (Granet-Abisset, 2016) from the 14th to the 18th century. This area was notable for having been able to impose a charter of freedom on the seigniorial power, allowing more self-governing than in other valleys with a tax-raising organization (Prost, 2004). Furthermore, The Grand Escarton territories had their own local management, by 'Communauté': the Vallouise Communauté, included Vallouise, Pelvoux, Les Vigneaux and Puy-Saint Vincent municipalities until the French revolution. As a result, numerous archives related to the management of the 'Communauté' were produced (Prost, 2005).

Until the beginning of tourism, Vallouise-Pelvoux economy was based mainly on agropastoralism (Laslaz, 2006). The rise of tourism towards the end of the 19th century, the expansion of means of communication and transport and the emergence of a strong diversification of mountain sports activities, especially in summer, led to significant changes, particularly in terms of visitor numbers, land use, and type of economic activities (Thénoz, 1981). There are few quantitative reports on tourist numbers in the municipality. However, the number of overnight stays at two campsites rose from 58,990 in 1968 to 131,661 in 1976. The same applies to the Glacier Blanc and Pelvoux refuges, which rose from 1,449 overnight stays in 1946 to 23,020 in 1975. As a result, tourist accommodations and leisure facilities have expanded to accommodate the flow of visitors, from 101 secondary houses or dwellings with temporary occupation in 1968, versus 1,588 in 2021 (INSEE, 2021). These changes have led to a decline in agriculture and a recovery in forestry: in 1856, 6 % of the municipality was covered by forests, compared with 22 % in 2018 (Defernand, 2021). Today, Vallouise-Pelvoux is well-known for its skiable domain and recognised as a major centre for mountaineering, kayaking and other outdoor sports. In terms of population, the municipality dropped from 1832 inhabitants in 1800 to 688 in 1962 following agriculture abandonment, but re-rose to 1132 in 2021. The urbanization of Vallouise-Pelvoux is concentrated in the floor of the valley at Village-Vallouise, but also includes a number of hamlets (most of which are inhabited year-round) that were once mountain pasture hamlets. Today, the urban fabric is more continuous than in the past, and the hamlets are linked by a denser network of local and departmental roads.

These various socio-economic factors generate risks and therefore damage to the various elements at risk present in the area. Buildings and communication networks are concentrated mainly in valley bottoms along rivers, and are therefore subject to damage by flooding, but also by avalanches, rockfalls (Figure 1) and landslides from overhanging slopes. Mountain infrastructures and activities (footpaths, ski resorts, etc.) spread out over the municipality wide range of elevations are also exposed to these hydrological and gravitational hazards.

## 3 Material and Methods

### 3.1 Overview of sources

Our methodology aimed at obtaining a multirisk event database as comprehensive as possible, combining the compilation of different pre-existing sources, and new archival searches. Therefore, the first step was to compile the information contained in existing databases produced by French public services, regulatory documents and the literature (Sect. 3.2). Then, our inventory was completed by archival sources not yet researched, or at least not exhaustively. The archival research strategy was based on a general analysis of i) the territory, ii) the relevant elements at risks that could be impacted or damaged by hazards, and iii) the actors or administrative bodies involved in risk management. As a result, three main categories of archives were considered, associated with the different territorial administrations, especially those managing forests, water and roads:

i) the municipal archives of Vallouise-Pelvoux include various documents related to the management of the municipality. They provide information dating back to the end of the 19th century at the latest (Following French archive management practices, all older archives have been deposited and kept in the Hautes-Alpes departmental archives).

ii) the departmental archives essentially in the Hautes-Alpes, with the exception of subseries 2C. Indeed, due to the turbulent

history of this region (see Sect. 2.2), some of the Dauphiné Intendancy (equivalent of the prefecture before the French revolution) archives are held by the Isère departmental archives.

iii) the Mountain Land Restoration service (RTM-ONF) archives, which contain documents related to the RTM "event" database called BDRTM, and other types of documents in relation to repair work carried out after events, or work on defence structures.

The variety of pre-existing and new sources mobilized in the constitution of the database allowed us to cover the 420-year study period. Figure 2 presents the time range potentially covered by the different sources. This, however, does not mean that information and events have been found on the full illustrated period for each source. Three of the pre-existing sources cover the totality of the period studied, namely databases that combine compilation of recent events (BDRTM and BDMVT) and previous research in archival sources. The series and sub-series consulted at the departmental archives were sought to cover

the oldest periods. The municipal archives deposited at the departmental archives of both municipalities cover the period 1600-1950, while the communal archives cover the 1950-2020 period. The C series from both departmental archives (Hautes-Alpes and Isère) also cover the oldest period (1600-1790), followed by the L series (1790-1800) and, the O, P and S series (1800-1940). Note that letters such as C, L, O, etc. correspond to the coding system used by the French archives and they do not have an extra specific meaning. Figure 2 and Table 2 give the full name of series and subseries, which are further detailed in Sect.

3.3. The RTM-ONF archives contain documents dating from the 1920-2020 period, older documents being likely deposited in the departmental archives.

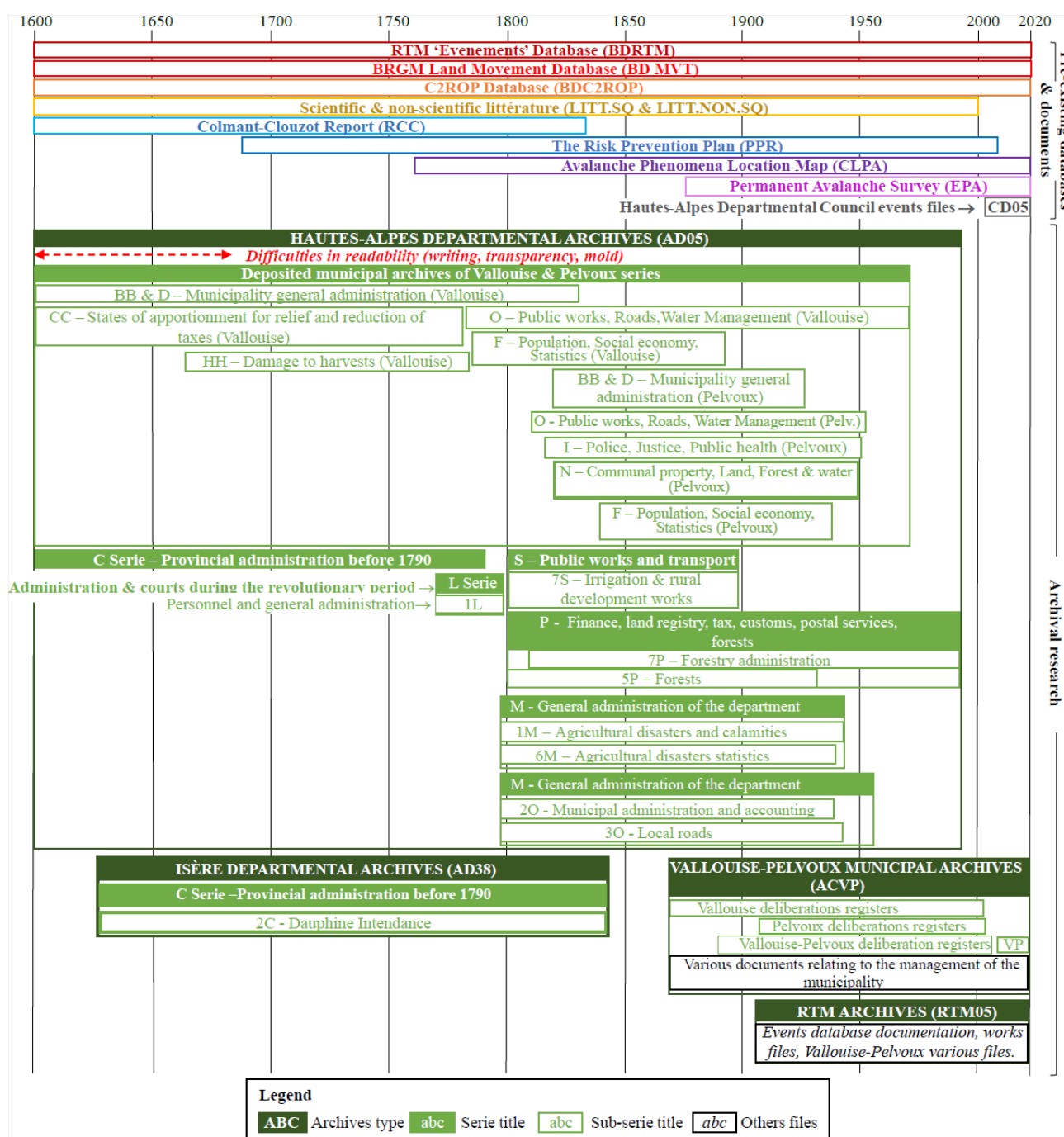

**Figure 2: Data used for the Vallouise-Pelvoux multirisk database. The time periods illustrated represent the period on which the documents contained in the series/sub-series were produced, not the periods over which information relating to natural hazards that occurred in the Vallouise-Pelvoux municipality was recorded. Pre-existing databases and documents detailed in Table 1 appear on top of the figure, and additional archival sources are listed below in green. Each series and sub-series is detailed by its reference and title. Letters such as C, L, O, etc. correspond to the coding system used by the French archives and they do not have an extra specific meaning. Figure 2 and Table 2 give the full name of series and subseries, which are further detailed in Sect. 3.3. The series and sub-series of the departmental archives, their titles and time periods, are further detailed in Table 2.**

## 3.2 Pre-existing databases and documents

Nine different pre-existing databases and documents have contributed to our database (Table 1).

| Name | Temporal period | Producer | Observed Phenomenon | Data type |
|---|---|---|---|---|
| **BDRTM** <br> *'Evenements' Database* | 563 - 2025 | ONF-RTM services | Hydrological & gravitational | Database & event sheets |
| **EPA** <br> *Enquête Permanente des Avalanches* | 1892 - 2025 | INRAE | Avalanches | Database |
| **BDMVT** <br> *The Land Movement database* | IIIth century - 2024 | BRGM | Landslides, rockfalls, collapses, mudslides, bank erosion | Database |
| **BDC2ROP** <br> *C2ROP event database* | 1191 - 2025 | INRAE | Rockfalls | Database |
| **RCC** <br> *Colmant-Clouzot report* | 1058 - 1843 | Pierre Colmant & Etienne Clouzot | All natural & climatic phenomenon | Working documents – archive copies |
| **CLPA** <br> *Avalanche phenomena location map* | 1400 - 2025 | INRAE | Avalanches | Maps & data sheets |
| **PPR** <br> *Risk Prevention Plan* | 1681 - 2004 | The French State | Floods, rockfalls & avalanches | Statutory report |
| **CD05** <br> *Event sheets* | 2008 - 2025 | Hautes-Alpes Department | Rockfalls, landslides, avalanches, floods | Event sheets |
| **SCIENTIFIC & NON-SCIENTIFIC LITERATURE** | 1422 - 1987 | Various Authors | All phenomenon | Books, thesis & articles |

**Table 1: Pre-existing databases and documents used to build the Vallouise-Pelvoux multirisk database and their various characteristics: full name/title, date of the oldest and newest mention, main data producers, hazards concerned and nature of the data.**

### BD RTM (563 to 2024)

The 'Evenements' Database is produced by the Office National des Forêts (ONF), specifically by the mountain land restoration service (RTM). This multirisk database records events related to natural hazards (torrential flooding, landslides, rockfalls, avalanches, etc.) that have occurred in mountain areas (12 departments in the French Alps and Pyrenees), as well as the protective structures present in these areas. The events in the BDRTM originate from regular live observations, but also stem from occasional research into historical hazards for the production of local risk zoning plans (PPRs) or the characterization of hydrological regime or the design of engineering structures (Bisquert et al., 2025). Blanchard et al. (2006) carried out historical researches to enrich the database for the most ancient periods for the Hautes-Alpes department.

### EPA (1892 to 2024)

The EPA (*Enquête Permanente des Avalanches*) is a regular system for observing and monitoring avalanches in France produced by the Office National des Forêts (ONF), resulting in a database managed by INRAE obtained by systematically recording observed avalanches on predefined sites (paths) (Bourova et al., 2016). It features a wealth of information on spontaneous avalanche activity in the French Alps and Pyrenees, useful for assessing spatio-temporal variability and related risks (e.g. Eckert et al., 2010; Lavigne et al., 2015). The EPA sites were originally chosen on the basis of forest damage. Today, other criteria such as elements at risk and scientific interest are favoured.

### BD MVT (3rd century to 2024)

The Land Movement database of the *Bureau de Recherches Géologiques et Minières* (BRGM; Geological and Mining Research Bureau) records various gravitational hazards including rockfall and landslides. It is a database and a map open to the public, which provides information on the characteristics of hazards and the associated damage. The data originate from older local databases, archives, partial inventories held by the organisations and a variety of other sources (media, studies, individuals, local authorities, associations, etc.).

### BDC2ROP (1191 to 2024)

The C2ROP rockfall event database was developed as part of the national C2ROP project, which, since 2014, has brought together operational and academic players working on rockfalls hazards (Eckert et al., 2020). This database gathered the pre-existing data available from public and private actors on rockfalls, their characteristics and damage caused in the French Alps.

*CLPA (Since 1400 to 2024)*

The *Carte de Localisation des Phénomènes d'Avalanche* (Avalanche phenomena location map) is an informative cartographic document that spatially describes maximal avalanche extents in the French Alps and Pyrenees. This map is the product of aerial photo studies, field investigations and the collection of testimonies through surveys. Each avalanche zone has a data sheet at the origin of the map and provides information on past avalanche events (Bonnefoy et al., 2010; Bourova et al., 2016). Informative documents used to build the map such as testimonies have been used for this study.

*RCC (1058 to 1843)*

The Colmant-Clouzot unpublished report is a collection of working documents, mainly copies of archival texts from several municipalities of the Dauphiné (Isère and Hautes-Alpes French departments) compiled by the archivist-palaeographers Pierre Colmant in 1907 and Etienne Clouzot who completed the work in 1913. This report was initially developed to gather past information concerning climatic and glaciological conditions and various related phenomena such as hazards, in order to study their variations over long periods. It takes the form of index cards, which are summary copies of various archive documents (available from the Hautes-Alpes and Isère Departmental Archives), resulting in a substantial collection of information on floods, landslides, avalanches, storm, frost spells and other climatological phenomena. The amount of information relating to glaciers, however, was more limited. This work was never published though some part was used by Mougin (1931) in his thesaurus book 'La Restauration des Alpes'.

*PPR (1681-2004)*

The Risk Prevention Plan is a French regulatory document that aims to identify, map and prevent the risks existing in municipalities. It is produced under the authority of the prefect of the department concerned and it defines the geographical areas exposed to natural hazards and the related risk levels. This document also prescribes measures and regulations to protect exposed populations. The types of hazards analysed in an area covered by a PPR are based on the analysis of a selection of historical data on past events, collected from the departmental archives and referenced in the document (Pigeon, 2007).

*CD05 (2008 to 2024)*

Documents produced by the Hautes-Alpes department which is responsible for road maintenance were used. These documents are descriptive sheets of events that caused damage to the road network in the Hautes-Alpes department. They provide information on the hazards, especially on the damage, the resulting interventions of the services and/or the work that was needed to solve the issues.

*Scientific and non-scientific literature (1427 to 1987)*

Various published documents provide information on past events in Vallouise-Pelvoux. Published scientific works such as the doctoral theses by Colas (2000) and Baraille (2001), or Mougin (1931)'s book on the restoration of the Alps, which listed some events identified in the Colmant-Clouzot report. Other non-scientific literature, is focused on the presentation and history of the valley (Séranon, 1891; Han, 1977; Cézard, 1981), provides an occasional source of information on the hazards in the area. This literature describes various hazards such as avalanches, rockfall and floods, and, for certain events, provides information on the assets at stake.

**3.3 Archival searches**

This first set of pre-existing sources used (sect. 3.2), although substantial, presents weaknesses. First, most of the sources relate to only one type of hazard (EPA, CLPA, BDC2ROP) or to one family of hazards (BDMVT). Second, with the exception of the EPA, the sources are not produced using a systematic survey, and mainly include events that have caused damage, leaving aside hazards that have occurred without any effect on the community. Finally, none of the sources provide information on glacial hazards, and we notice a lack of documentation for the oldest periods (1600-1900) which are very important to analyze hazard evolution through time and space.

To compensate for these shortcomings, archival research was carried out using the method presented in Sect. 3.1. Most of the archives consulted were in the Hautes-Alpes departmental archives, which hold documents on Vallouise-Pelvoux dating from the 17th century to the present day. The series consulted, presented in Table 2, are closely linked to the administrative
management of the area and the vulnerability of the community to natural hazards.

The first focus was on the communal archives deposited in the departmental archives to supplement the research carried out at the Vallouise-Pelvoux municipal archives (Table 2). The subseries likely to contain information are the registers of deliberations, which list municipal decisions in all areas of municipal management. There are also subseries related to crop damage and tax exemptions, which were granted to compensate residents for the damage caused by hazards. Finally, resources
related to constructive works, roads and water. Other series reviewed, were produced by other administrations including series related to the provincial (C), the revolutionary period (L) and the department (M) Administrations. Secondly, following the same pattern as the subseries of the deposited archives, we focused on public works, road, water management (O & S series) and on forestry (5P). The 'Eaux & Forêts' administration (7P) provided data on wood management, reforestation, wood cutting, dams, roads but also studies on torrents, floods and mountainous perimeters. The U series, linked to the Justice Department,
was also consulted for claims, declarations and commission files on agricultural or industrial damage.

| DEPARTEMENTAL ARCHIVES OF THE HAUTES-ALPES (AD05) | |
|---|---|
| **Deposited archives of Vallouise & Pelvoux** | **1346-1951** |
| *Deposited municipal archives of Pelvoux* | *1643-1951* |
| *Deposited municipal archives of Vallouise* | *1346-1890* |
| **C series - Provincial administration before 1790** | **1127-1793** |
| **L series - Administration & court during the revolutionary period** | **1789-1800** |
| *1L subseries - Administration holdings* | *1790-1800* |
| **M series - General administration of the Hautes-Alpes department** | **1799-1943** |
| *1M Subseries - General administration of the department* | *1799-1943* |
| *6M Subseries - Population - Economy - Statistic* | *1789-1940* |
| **O series - Public works, roads, water management** | **1800-1940** |
| *2O - Municipal administration and accounting* | *1800-1940* |
| *3O - Local roads* | *1793-1956* |
| **P series - Finance, Land Registry, Taxation, Customs, Post Office, Forest** | **1800-1940** |
| *5P subseries - Forests* | *Year VIII\*-1939* |
| *7P subseries - Forestry administration* | *1834-1988* |
| **S series - Public works and transport** | **1800-1940** |
| *7S subseries - Irrigation & rural development works* | *1763-1988* |
| **U series - Justice** | **Since 1800** |
| *3U subseries - Courts of First Instance* | *1800-1958* |
| DEPARTEMENTAL ARCHIVES OF ISERE (AD38) | |
| **C series - Provincial administration before 1790** | **1241-XIXe** |
| *2C subseries - Dauphine Intendance* | 1241-1792 |

*\*Year VIII = this revolutionary year began on September 23rd, 1799 and ended on September 22nd, 1800.*

**Table 2: Series and sub-series browsed in the Isere and Hautes-Alpes departmental archives and their temporal coverage. The series and sub-series concern the various administrative units to which Vallouise-Pelvoux belongs: Le Dauphiné, Hautes-Alpes department, the municipality, as well as the various elements at risk and territorial units in charge of their management that may
be affected by hazards: roads, forests, water management, etc. The temporal coverage of these series and sub-series is displayed in Figure 2.**

Data gathered from these archival searches comes from a variety of document types. Some difficulties have been encountered, particularly concerning old archives from 1600 to 1680. Their readability was sometimes limited because of the writing, the transparency of the paper, the wear and tear of time or ink, or even the mould present. Any document found to contain
information useful for compiling the database was collected (photocopies or photographs) and transcribed as shown in Fig. 3. Transcription of the documents was then used to fill in and code the database (Sect. 3.4). Deliberations, taken from the registers of deliberations, are the unique type of document available over the entire period studied. Otherwise, the documents are extremely varied: letters, reports, studies, event registration forms, municipal by-laws, statistical tables, requests for wood to be felled, etc. (Figure 3).

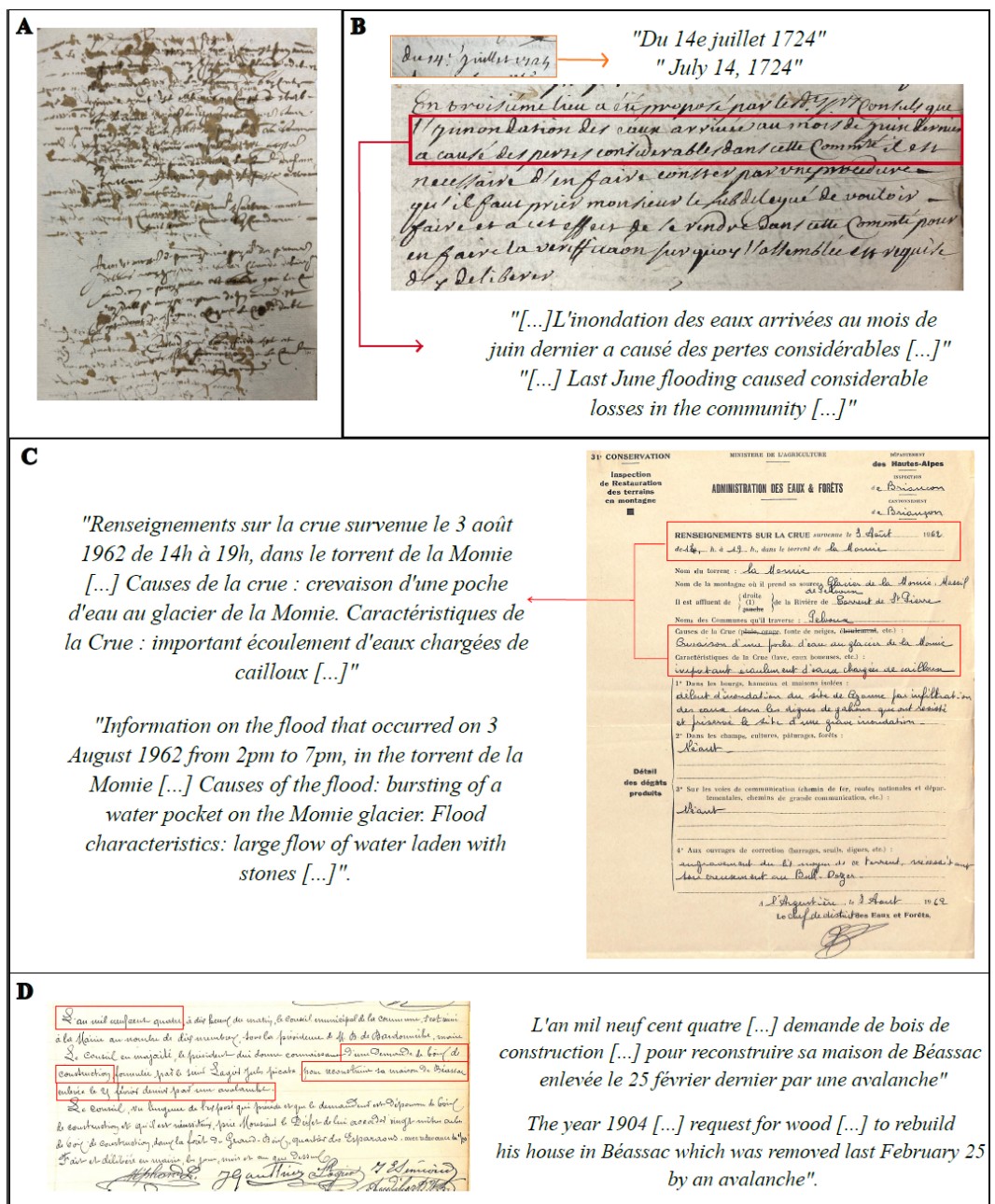

**Figure 3: Historical sources used to populate the Vallouise-Pelvoux multirisk database. A - Example of a municipal deliberation archival source that is difficult to decipher due to the shape of the writing, the transparency of the paper and the ink stains, consulted at the Hautes-Alpes departmental archives (Deposited municipal archive of Vallouise – 1668-1669 - BB30). B - Example of a deliberation on July 17, 1724, mentioning a flood that caused damage in Vallouise-Pelvoux, which was not recorded in pre-existing databases and documents (source: Hautes Alpes departmental archives in the Vallouise deposited communal archives series, BB41). C- Example of an event sheets from the ONF-RTM archives mentioning a flood resulting from the rupture of a glacial water pocket (source: Hautes Alpes departmental archives in the Vallouise deposited communal archives series BB41). D - Example of a municipal deliberation in 1904, mentioning an avalanche (source: Vallouise-Pelvoux Municipal Archives, deliberations register). Red boxes were drawn to highlight the segments of original text for which a translation is provided.**

### 3.4 Database characteristics

Our multirisk database compiles data on the nature of the events, their impacts and the corpus of sources used to justify the inventoried information, so that a variety of analyses can be carried out on different themes. The structure of the database along with definitions is given below. It is important to note that the database was built with the objective to avoid any subjectivity, restricting strictly the information recorded to what the sources actually say, regarding, e.g. the nature of the hazard and/or interactions between hazards and/or impacts.

### 3.4.1 Hazard

The multirisk database gathers a range of information to characterize various hazards and their impacts over a pluricentennial time frame: floods, rockfalls, avalanches, landslides and glacial hazards. For floods, when sources distinguish "standard" floods (more or less concentrated in sediments) from debris flows, this information was kept, but this was mostly possible from the early 20th century onward only, as debris flows were generally not recognised as specific processes earlier, see discussion. For rockfalls hazards, when known, categories relating to the volume of the rockfalls have been added: isolated boulder, several boulders and large rock avalanche, as well as the total volume. The database also lists cascading multihazards that could be identified from sources. These appear as distinct pairs of events in the database, because the nature of the phenomenon and its location vary, but information on the causal link between them is kept.

### 3.4.2 Event

The event is the primary entry of the database, defined first and foremost by its site, i.e., the place where it occurred: avalanche paths, rivers and torrents, and precise spatial coordinates for rockfall and landslides. This choice results in the potential existence of several events for the same hazard, at the same date, particularly for flood episodes. Indeed, there are several cases where events are linked on different scales: events of the same hazard that have no direct link, apart from the weather (e.g., Avalanches episodes); events of the same hazard that are partly linked (e.g., floods on different torrents); or cascading multihazard events. When the precise location of an event is unknown, it is automatically classified within the administrative boundaries of the municipality. For each event, other key information is the date on which the event occurred, related sources, hazard type and impacts, if any.

### 3.4.3 Sources

Sources related to events were classified in 14 categories. The category refers to the pre-existing or archival material. The datasets contain the references of the documents testifying the event, the place where they are conserved, the type of document (e.g., letter, report, photography, *etc.*), the date of the documents as well as a verbatim transcript.

### 3.4.4 Assets at risk affected, damaged or destroyed

Assets at risk affected, damaged or destroyed by hazards are recorded in the database in a comprehensive way. The categorization of the assets is organized into four main categories: material, functional, human and environmental. Each of these categories are subdivided into a total of 19 sub-categories. Material assets include residential buildings, infrastructures, economical buildings, defence structures, vehicles. Functional assets cover roads, trails, networks (electric, water, telephone) and crossing structures such as bridges. Human assets record any evacuated, injured, dead or unharmed people and their number. The environmental category lists the impacts on agricultural land, forests, and rivers. Material, functional and environmental categories have an 'others' subcategory. Each of the sub-categories of material, functional and environmental damage has its own gradation which specifies the nature and/or extent of the damage caused. Each material subcategories can be detailed as reached, damaged or destroyed. The functional sub-categories have a 'partial obstruction' or 'total obstruction' gradation. Environmental subcategories have more specific gradations such as 'erosion', 'deposition', 'change of bed' for rivers or 'flooded', 'washed away', 'deepened', 'hollowed out' or 'gullied' for agricultural land. Details are also provided on the certainty of the damage: uncertainty of damage concerns events taking place at the same time on different sites, but no exact location in the sources. For each event, as many impacts as necessary can be documented, allowing, e.g., exploration of cascading consequences.

**4 Results**

In Sect. 4.1, the 3,528 hazard mentions gathered from the various sources investigated are presented. Since different mentions may refer to the same hazard and impacts, the whole corpus results in 2,131 single events introduced in Sect. 4.2. Sect 4.3 describes the overall characteristics of these events, while Sect.4.4. focuses on their various impacts and Sect. 4.5 on their temporal distribution.

**4.1 Contribution of the corpus of sources**

The same event can be mentioned several times and/or in several categories of the corpus of sources, so the number of mentions and events should not be confused. In total, the corpus of sources provided a total of 3,528 mentions (Fig. 4). The biggest contributor is the EPA, with 1,470 mentions of avalanches, followed by the Hautes-Alpes Departmental Archives with 1,184 mentions. The categories with the lowest contributions are the BDMVT and the Hautes-Alpes Departmental Council data sheets. The compilation from pre-existing sources has produced 2,084 mentions, while the archival searches have resulted in

the recording of 1,444 new mentions.

The 3,528 mentions concern mostly avalanches (n=2654), floods (n=762, including debris flows) and rockfall (n=88). By contrast, landslides (n=15) and glacial hazards (n=9) represent lower proportions of mentions (Fig. 4B). The over representation of avalanches in sources results from their systematic survey on 59 paths of the municipality since 1919 within the EPA, and the previous form of the systematic survey that existed since the end of the 19th century: the avalanche

identification booklet, preserved in the departmental archives. As expected, the largest amount of information on rockfall comes from the BDC2ROP, which focuses on this hazard. The Hautes-Alpes Departmental Archives provided information mainly on avalanches, while the BDRTM provided more information on floods. The 9 glacial hazards are only represented in a single source: the documentation of the ONF-RTM services. 19 mentions concern multihazard cascading events: 9 in the ONF-RTM archives, 5 in the BDRTM, 2 in the CLPA, and mention one in the AD05, the EPA and the non-scientific literature

(LITT NON SQ), respectively.

The corpus of sources mobilised covers different time periods. Their diversity enables us to cover the entire 1600-2020 period. In comparison with Fig. 2, Fig. 4A shows the temporal period over which mentions have been recorded for each source in the database. Events within our study area are over a shorter period than at the national scale especially for the BDMVT, the BDC2ROP and the CLPA.

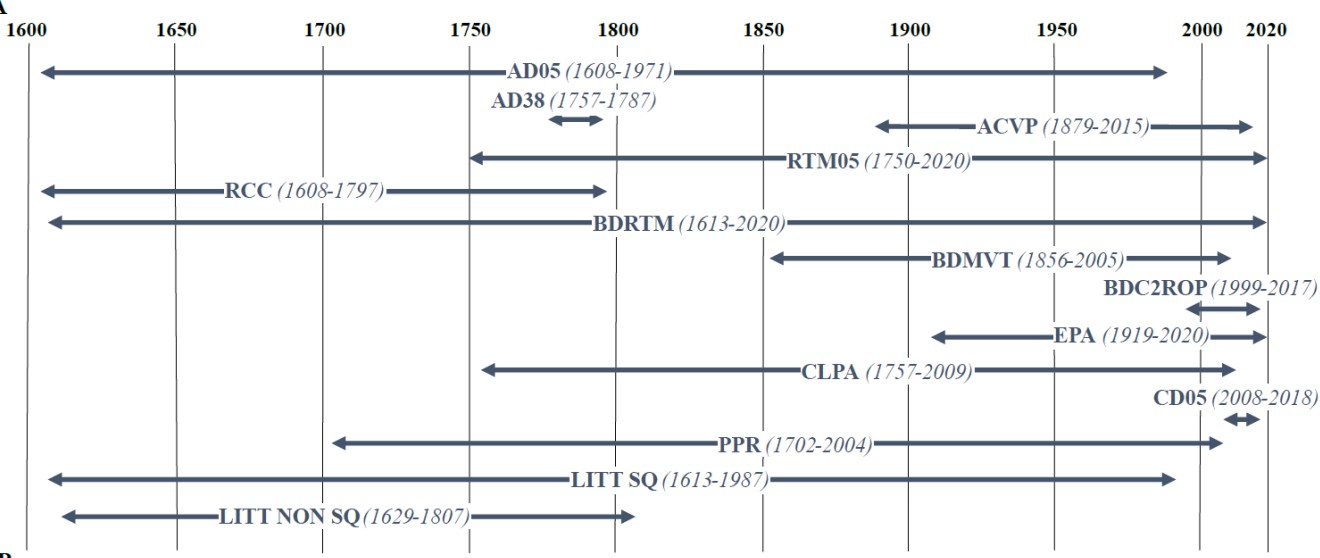

**B**

| SOURCES | AVALANCHES | FLOODS | ROCKFALLS | LANDSLIDES | GLACIAL HAZARD | TOTAL |
|---|---|---|---|---|---|---|
| AD05 *(1608-1971)* | 988 | 189 | 7 | 0 | 0 | **1184** *(33.6%)* |
| AD38 *(1757-1787)* | 12 | 8 | 0 | 0 | 0 | **20** *(0.6%)* |
| ACVP *(1879-2015)* | 5 | 94 | 3 | 2 | 0 | **104** *(3%)* |
| RTM05 *(1750-2020)* | 12 | 98 | 13 | 4 | 9 | **136** *(3.8%)* |
| RCC *(1608-1797)* | 20 | 69 | 1 | 0 | 0 | **90** *(2.6%)* |
| BDRTM *(1613-2020)* | 51 | 212 | 14 | 5 | 0 | **282** *(8%)* |
| BDMVT *(1856-2005)* | 0 | 0 | 12 | 3 | 0 | **15** *(0.4%)* |
| BDC2ROP *(1999-2017)* | 0 | 1 | 22 | 0 | 0 | **23** *(0.7%)* |
| EPA *(1919-2020)* | 1470 | 0 | 0 | 0 | 0 | **1470** *(41.7%)* |
| CLPA *(1757-2009)* | 25 | 1 | 0 | 0 | 0 | **26** *(0.7%)* |
| CD05 *(2008-2018)* | 1 | 2 | 9 | 1 | 0 | **13** *(0.4%)* |
| PPR *(1702-2004)* | 48 | 8 | 5 | 0 | 0 | **61** *(1.7%)* |
| LITT SQ *(1613-1987)* | 18 | 66 | 2 | 0 | 0 | **86** *(2.4%)* |
| LITT NON SQ *(1629-1807)* | 4 | 14 | 0 | 0 | 0 | **18** *(0.5%)* |
| **TOTAL** | **2654** *(75.3%)* | **762** *(21.5%)* | **88** *(2.5%)* | **15** *(0.4%)* | **9** *(0.3%)* | **3528** *(100%)* |

**Figure 4: Statistical distribution of the mentions used to create the Vallouise-Pelvoux multirisk database (total n=3528). A – Their time frame within the different sources. B – Their number (and percentage with regards to the total number of mentions) as function of the different hazards. Under each source name is indicated the time period over which the sources provided events. The flood category includes all types of floods (more or less concentrated in sediments) and debris flows.**

### 4.2 From mentions to the event database

The 3,528 mentions in the corpus of sources have resulted in the creation of an event database containing 2,131 events: 1,654 avalanches, 418 floods (including debris flows), 44 rockfalls, 9 glacial hazards and 6 landslides (Table 3). Of these, 12 are cascading multihazard events, namely 6 pairs of events linked by a causal connection. Specifically, there are 4 pairs of events linking glacial hazards and flooding, one pair of events linking two floods (a debris flow as identified by the source and a flood in the strict sense of the term) and one pair of events linking an avalanche with a flood.

About half of the events are mentioned in two different sources (n=1,026), while just under half of the events (936 events) could only be found in a single source. The intersection of three sources is only found for 138 events, mostly floods with 71 events, while there are only 31 events mentioned in four or more sources. Rockfalls and glacial hazards events are for the largest proportion found in only one source. Floods, avalanche events are mainly originating from two sources.

| | AVALANCHES | FLOODS | ROCKFALLS | GLACIAL HAZARDS | LANDSLIDES | TOTAL |
|---|---|---|---|---|---|---|
| **Number of events** | 1654 | 418 | 44 | 9 | 6 | 2131 |
| Recorded in one source | 737 | 171 | 19 | 8 | 1 | **936** *(44%)* |
| Recorded in two sources | 848 | 163 | 12 | 1 | 2 | **1026** *(48%)* |
| Recorded in three sources | 58 | 71 | 7 | 0 | 2 | **138** *(7%)* |
| Recorded in four or more sources | 11 | 13 | 6 | 0 | 1 | **31** *(1%)* |

**Table 3: Number of events in the Vallouise-Pelvoux multirisk database as function of the number of sources in which they were reported (total n=2,131), in total and by hazard. Among the 12 multihazard cascading events, 6 are mentioned in one source, 5 in two sources and one in three sources. The flood category includes all types of floods (more or less concentrated in sediments) and**
415 **debris flows.**

**4.3 Statistical overview of the events**

The events in our database can be distinguished in three different classes (Figure 5A):

- New events resulting from the additional archives research;

- Events in pre-existing databases verified and/or corrected by the additional archival research;

- Events in pre-existing sources not found in the additional archives research.

Additional research enabled the inclusion in the database of new events that had not been compiled in pre-existing sources: 304 new events were added, 14 % of the database. Archival research was also used to verify and correct information about events compiled in existing sources, which means that the accuracy of information contained in pre-existing databases has been verified, corrected or supplemented by archival sources. A total of 1,089 events were verified and/or corrected
representing about half of our database (51 %). The remaining 738 events were derived solely from pre-existing sources and could not be verified by archive searches, representing 35 % of the database.

**A**

| | AVALANCHES | FLOODS | ROCKFALLS | LANDSLIDES | GLACIAL HAZARDS | TOTAL |
|---|---|---|---|---|---|---|
| New events | 137 | 148 | 10 | 0 | 9 | **304** *(14%)* |
| Event verified/corrected | 876 | 195 | 13 | 5 | 0 | **1089** *(51%)* |
| Events only found in pre-existing sources | 641 | 75 | 21 | 1 | 0 | **738** *(35%)* |

**B**

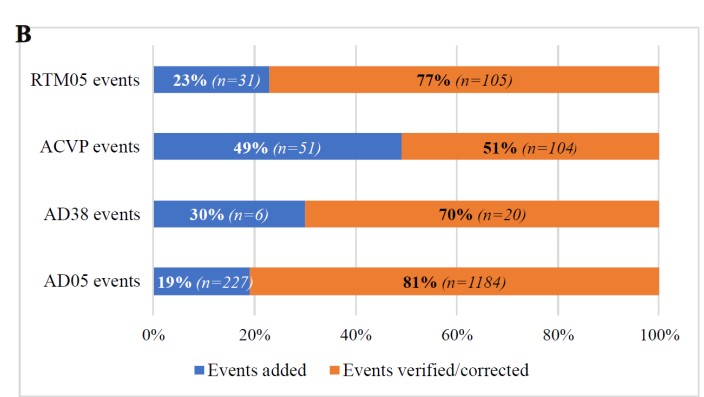

**Figure 5: Types of events by archive types and the contribution of the archival research to the constitution of the Vallouise-Pelvoux multirisk database. A - Number of events per hazard (n=2131) distinguishing those added from archival research; those pre-existing**
**that could be verified and/or corrected by the archival research; and those only found in pre-existing sources. B - Number and relative proportion of events added and verified/corrected as function of the archival source. Among the 12 multihazard cascading events, 10 come from the archives (9 in the RTM05 and one in the AD05), and 7 from pre-existing sources. The flood category includes all types of floods (more or less concentrated in sediments) and debris flows.**

Floods in all their diversity (including debris flows) are the most represented hazard in the new events with 148 events,

followed closely by avalanches with 137 events. Floods are even more represented in the new events in proportion: 148 events (46 %), against 137 events (8 %) for avalanches. A total of 10 rockfall new events has been added, as well as all 9 glacial hazards, but no new landslides. The 12 multihazard cascading events result in 5 new events (4 glacial hazards and one flood), 5 verified/corrected events (4 floods and one avalanche), leaving only two floods that are only found in pre-existing sources. Most of the events verified/corrected were avalanches (n=876), followed by 195 floods, 13 rockfalls and 5 landslides. If we

consider the proportion of contribution made by each type of archival source (Fig. 5B), 70 to 80 % of the events recorded were useful to verify/correct events from pre-existing sources. The Vallouise-Pelvoux Municipal archives (ACVP) made the largest contribution to new events, with half of the events recorded as new ones.

## 4.4 Statistical overview of damage

Overall, 1,069 events caused damage, while 967 caused no damage, and no information on damage was available for 95 events.

Table 4 synthesizes damage as function of hazards, depending on the accuracy and certainty of the data collected. The hazard with most damageable events is avalanches, followed by floods, rockfalls, landslides and then glacial hazards. The multihazard cascading events all produced damage, but 4 of them have no detailed information about it. Proportionally (i.e., with regards to the total number of events for a given hazard). However, the most events causing some level of damage are floods, followed by rockfall, landslides and avalanches. In fact, floods were recorded most of the time when they caused damage whereas

avalanches have been systematically recorded since 1919 for a given number of avalanche paths (EPA). The majority of the damaging events in the database provide detailed information on the damage caused (89 %) and 95 % of these detailed damage have a precise location. Floods present the highest level of uncertainty in terms of exact location of the damage caused due to their often-widespread extent.

| | AVALANCHES | FLOODS | ROCKFALLS | GLACIAL HAZARDS | LANDSLIDES | TOTAL |
|---|---|---|---|---|---|---|
| Number of events | 1654 | 418 | 44 | 9 | 6 | 2131 |
| Events with damage | 634 | 393 | 34 | 5 | 3 | 1069 (50%) |
| Events without damage | 965 | 1 | 1 | 0 | 0 | 967 (45%) |
| Events with no information about damage | 55 | 24 | 9 | 4 | 3 | 95 (5%) |
| With detailed damage | 587 | 325 | 31 | 1 | 3 | 947 (89%) |
| Without detailed damage | 47 | 68 | 3 | 4 | 0 | 122 (11%) |
| With located detailed damage | 578 | 282 | 31 | 1 | 3 | 895 (95%) |
| With unlocated detailed damage | 9 | 43 | 0 | 0 | 0 | 52 (5%) |

**Table 4: Overall damage statistics for the events of the Vallouise-Pelvoux multirisk database. In total and for each hazard, the table provides the number of events with damage or where there is no information about damage, whether the damage information (if any) is detailed or not, and whether damage information is certain or not. Uncertainty regarding damage concerns events taking place at the same time on different sites, for which the information from the sources does not make it possible to know the exact location. The 12 cascading hazard events all produced damage, and we know the details of this damage for 8 of these events. The**
**flood category includes all types of floods (more or less concentrated in sediments) and debris flows.**

Damage produced have been classified into four major categories, and 19 sub-categories (Table 5). An event may have caused damage in several categories and several sub-categories. Indeed, for a given event, one to four damage categories can be

represented, with one to six sub-categories. For example, one of the multihazards of our database consisted of an avalanche that broke trees in its flow path. Snow and trunks were deposited in a river bed (environmental damage category, forest and watercourse damage subcategories), causing the river to overflow onto the road (material and functional damage categories, infrastructure and roads damage subcategories). Hence, cascading consequences are documented and their causal relations remain traced.

Of the 1,069 events that caused damage, there was a total of 1,674 damage recorded in the four major categories, including 563 environmental, 539 functional, 552 material and 20 human damage. The 'infrastructure' sub-category, which includes bridges and roads, is the one with the most damage (n=446), followed logically by the 'roads' sub-category (n=354). In the environmental damage category, the 'river' sub-category recorded the most damage (n=310), followed by the 'forest' (n=143) and 'agricultural land' (n=122) sub-categories. Note that the number of injured, dead and unharmed people may be lower than the number of events with human damage, because the number of people affected was only recorded when it was known with certainty.

| | AVALANCHES | FLOODS | ROCKFALLS | LANDSLIDES | GLACIAL HAZARDS | TOTAL |
|---|---|---|---|---|---|---|
| **EVENTS WITH MATERIAL DAMAGE** | **324** | **203** | **23** | **2** | **0** | **552** |
| **Residential buildings** | 43 | 17 | 5 | 0 | 0 | 65 |
| **Infrastructures** | 275 | 156 | 14 | 1 | 0 | 446 |
| **Economical buildings** | 9 | 13 | 0 | 0 | 0 | 22 |
| **Protective structures** | 3 | 57 | 1 | 0 | 0 | 61 |
| **Vehicles** | 0 | 4 | 2 | 0 | 0 | 6 |
| **Others** | 1 | 14 | 3 | 0 | 0 | 18 |
| **EVENTS WITH FUNCTIONNAL DAMAGE** | **358** | **162** | **17** | **2** | **0** | **539** |
| **Roads** | 262 | 79 | 12 | 1 | 0 | 354 |
| **Trails** | 91 | 43 | 3 | 0 | 0 | 137 |
| **Networks** | 2 | 18 | 1 | 1 | 0 | 22 |
| **Crossing structures** | 14 | 73 | 2 | 0 | 0 | 89 |
| **Others** | 0 | 3 | 0 | 0 | 0 | 3 |
| **EVENTS WITH HUMAN DAMAGE** | **6** | **11** | **1** | **1** | **1** | **20** |
| **Evacuation** | 0 | 11 | 0 | 1 | 1 | 13 |
| *Number of people evacuated* | 0 | 35 | 0 | 20 | 4 | 59 |
| **Injured** | 3 | 0 | 0 | 1 | 1 | 5 |
| *Number of injured people* | 1 | 0 | 0 | 3 | 3 | 7 |
| **Deaths** | 4 | 0 | 1 | 1 | 1 | 7 |
| *Number of deaths* | 48 | 0 | 1 | 4 | 1 | 54 |
| **Unharmed** | 2 | 0 | 0 | 1 | 0 | 3 |
| *Number of unharmed people* | 3 | 0 | 0 | 9 | 0 | 12 |
| **EVENTS WITH ENVIRONMENTAL DAMAGES** | **374** | **185** | **4** | **0** | **0** | **563** |
| **Agricultural land** | 5 | 117 | 0 | 0 | 0 | 122 |
| **Forests** | 126 | 16 | 1 | 0 | 0 | 143 |
| **Rivers** | 265 | 45 | 0 | 0 | 0 | 310 |
| **Others** | 3 | 29 | 3 | 0 | 0 | 35 |

**Table 5: Damage statistics for the events of the Vallouise-Pelvoux multirisk database. Classification into 4 main categories (in light blue): material, functional, human and environmental damage, and 19 sub-categories (in white). For human damage (cf. 3.4) the number of people injured, killed or uninjured is also provided. A single event may have produced damage from several categories and sub-categories. The 12 cascading multihazard events mainly include the damage sub-categories 'evacuation' (n=3, 23 people evacuated), 'infrastructure' (n=3) and 'watercourse' (n=3). The flood category includes all types of floods (more or less concentrated in sediments) and debris flows.**

## 4.5 Distribution of events over time

An interest of this database is to analyse the distribution of events over time. We below describe its shape and modulation as
function of sources and impacts, and Sect. 5.2 proposes first insights regarding its drivers.

Figure 6 illustrates the distribution of the events coming from the pre-existing sources and those added from archival research. The events mentioned in the pre-existing sources occurred mainly between 1920 and 2020. We can also note other rich periods, 1608-1640, 1750-1800 and 1850-1870, although less rich than the 1920-2020 period. The archival research has resulted in the input of events after 1670. The major period over which new searches in archives contribute to enrich the database is 1835-
2020, with a maximum number of new events over the period 1920-1970.

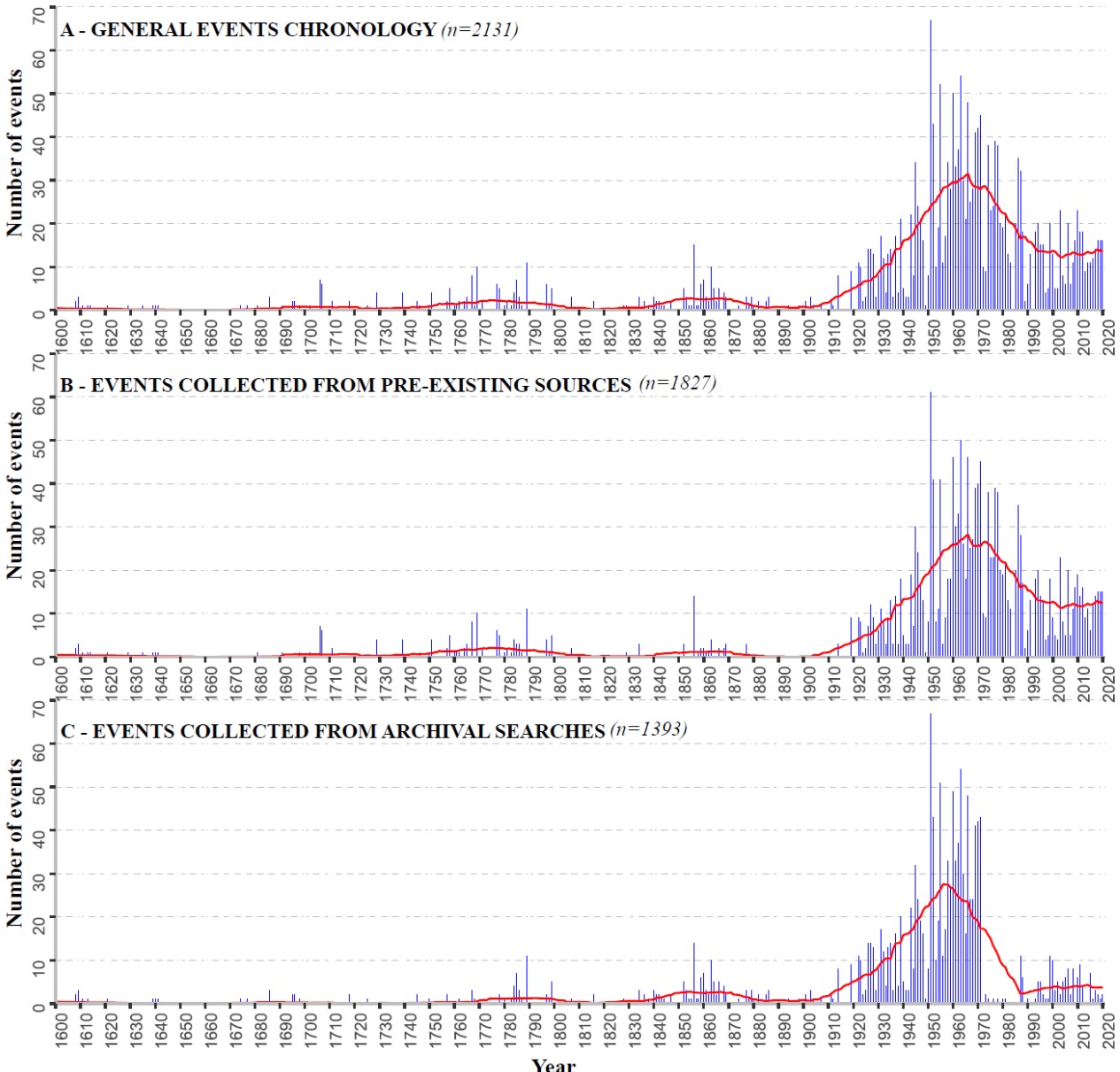

**Figure 6: 1600-2020 chronologies of the events from the Vallouise-Pelvoux multirisk database. (A) All events (B) Events from pre-existing sources and (C) event added from archives searches in departmental, municipal and ONF-RTM archives. Each temporal distribution is provided as raw annual counts with a 31-year moving average (red line). (A) graph is the intersection of**
**graphs (B) and (C), as some events in pre-existing sources were confirmed by the archival search.**

Among the biggest contributors to the database, the 1202 events recorded in the departmental archives range from 1608 to 1971, with the greatest contribution from the periods 1780-1800, 1840-1880 and 1900-1971. The 135 events from the municipal archives were recorded over a more recent period, from 1879 to 2015, with the largest numbers between 1926-1960

and 1980-2015. Finally, the archives of the ONF-RTM have revealed their 104 events over the period 1750-2020, concentrated mainly over the period 1930-2020 (Fig. 7).

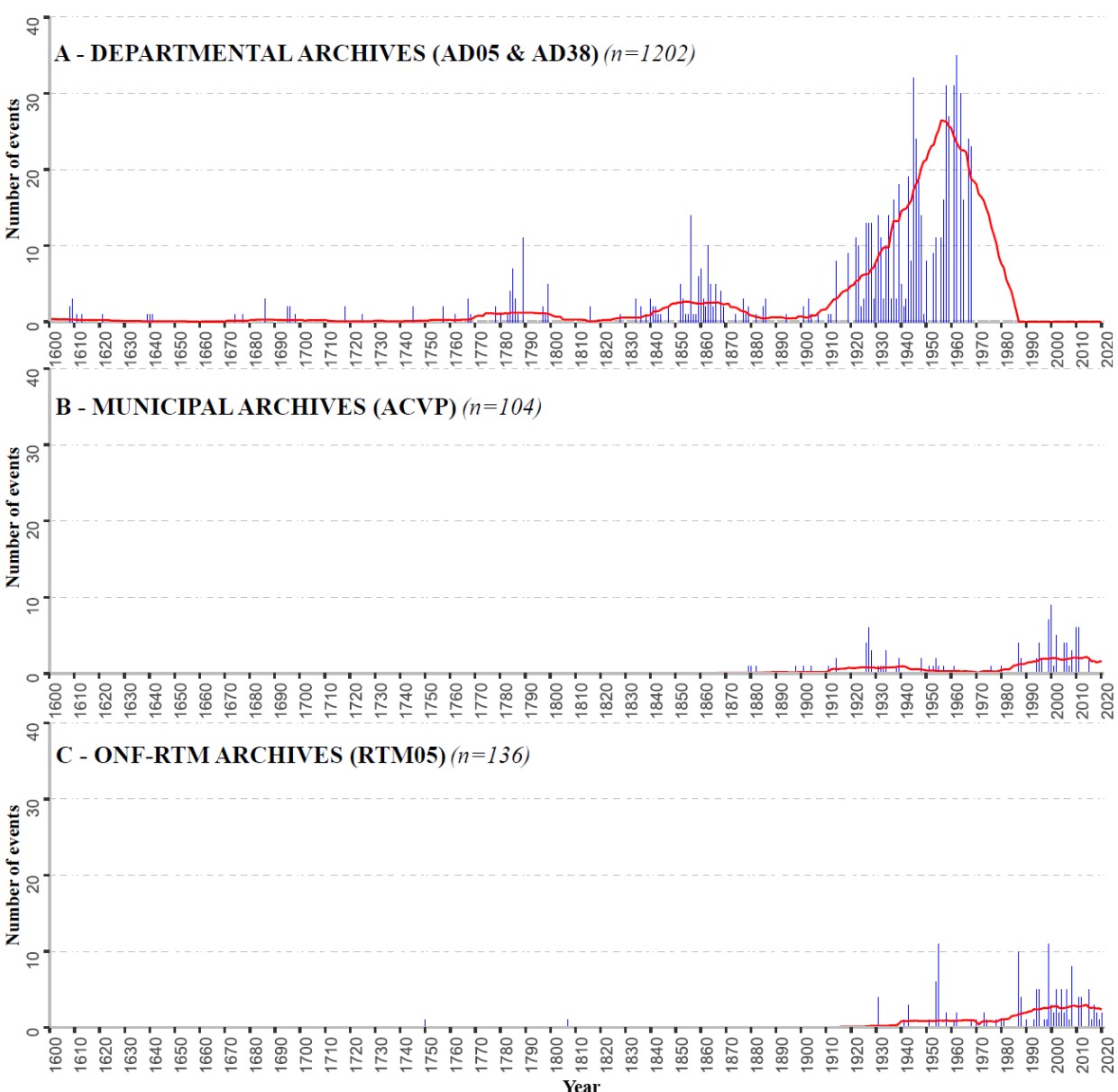

**Figure 7: 1600-2020 chronology of the events from the Vallouise-Pelvoux multirisk database retrieved from different archival sources. A- From departmental archives of the Hautes-Alpes and Isère (of which 4 % are communal archives deposited in the departmental archives of the Hautes-Alpes), B – From municipal archives of Vallouise-Pelvoux, C- From ONF-RTM archives. Each temporal distribution is provided as raw annual counts with a 31-year moving average. The sum of the three chronologies is equal to graph C in Figure 6.**

In Fig. 8A, we show the temporal distribution of the 304 'new' events resulting from research in the archives, and unknown in the pre-existing sources. The 'new' events are mainly related to event-rich periods, but occasionally 'new' events are added during oldest off-peak periods. Figure 8B shows that the 1,088 events first compiled in the pre-existing sources and found again in the archive research, cover mainly the period 1920-1970. Finally, Fig. 8C indicates that the events compiled from pre-existing sources but not found in the archives are mainly concentrated in the period 1970-2020.

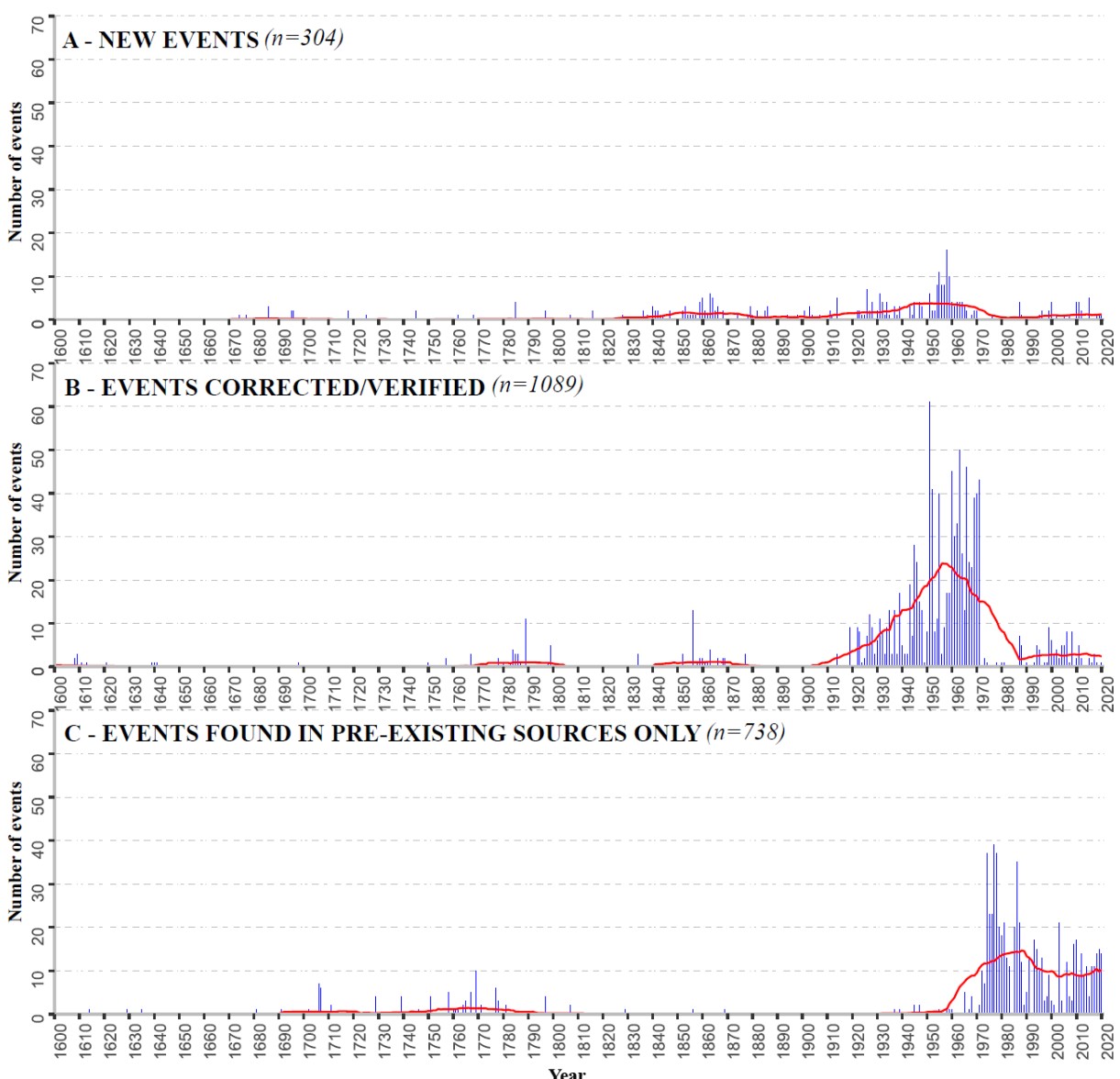

**Figure 8: 1600-2020 chronology of the events from the Vallouise-Pelvoux multirisk database highlighting: (A) New events added through archival research (n=304); (B) events compiled in pre-existing sources and retrieved through archival research, therefore cross-checked and/or corrected (n=1088); (C) events found in pre-existing sources only (n=737). Each temporal distribution is provided as raw annual counts and 31-year moving average.**

A total of 81 % (n=245) of the events added were events that caused damage (Fig. 9A): 140 floods, 91 avalanches, 9 rockfalls and 7 glacial hazards. The temporal distribution of these added events is mainly concentrated over the period 1919-2020 (64 %, n=195), versus 109 events added over the 1600-1919 period (36 %). However, the proportion of flood events added over the old period 1600-1919 is higher, with 82 floods added, i.e., 55 % of the total number of floods added. New added glacial hazards all correspond to multihazard cascading hazards, in 1809, 1958, 1962 and 2012, respectively.

| A | TOTAL | 1600-1919 | | 1920-2020 | |
|---|---|---|---|---|---|
| AVALANCHES | 137 | 21 | 15% | 116 | 85% |
| *Damaging events* | *91* | *21* | *23%* | *70* | *77%* |
| FLOODS | 148 | 82 | 55% | 66 | 45% |
| *Damaging events* | *140* | *80* | *57%* | *60* | *43%* |
| ROCKFALLS | 10 | 5 | 50% | 5 | 50% |
| *Damaging events* | *9* | *4* | *44%* | *5* | *56%* |
| LANSLIDES | 0 | 0 | 0% | 0 | 0% |
| *Damaging events* | *0* | *0* | *0%* | *0* | *0%* |
| GLACIAL HAZARDS | 9 | 1 | 11% | 8 | 89% |
| *Damaging events* | *5* | *1* | *20%* | *4* | *80%* |
| TOTAL | 304 | 109 (36%) | | 195 (64%) | |
| *Damaging events* | *245 (81%)* | *106 (43%)* | | *139 (57%)* | |

B

AVALANCHES: 8% — 92%
FLOODS: 35% — 65%
ROCKFALLS: 23% — 77%
LANDSLIDES: 100%
GLACIAL HAZARDS: 100%

0% 10% 20% 30% 40% 50% 60% 70% 80% 90% 100%

■ New events proportion   ■ Other events

**Figure 9: New events added to the database analysis (n=304); (A) showing the number of events per hazard and the proportion of damaging event for each of them over the period 1600-1919 and 1920-2020 and in total; (B) Proportion of new events per hazard. The flood category includes all types of floods (more or less concentrated in sediments) and debris flows.**

Avalanche and flood events (in all their diversity) are distributed over almost the entire study period: 1629-2020 for avalanches and 1608-2020 for floods (Table 6). The majority of avalanche events occurred over the last 100 years (97 %) as well as landslides (100 %), glacial hazards (89 %), rockfall (84 %). Events corresponding to rockfall, landslides and glacial hazards, on the other hand, are distributed over shorter, more recent periods: 1781-2017 for rockfalls, 1807-2020 for glacial hazards and 1941-2015 for landslides. The multihazard cascading events cover the period from 1807 to 2012, with 83 % of them over the last 100 years.

| | Oldest event | Latest event | Number of events | Number of event over the last 100 years |
|---|---|---|---|---|
| AVALANCHES | 1629 | 2020 | **1654** | **1587** *(96%)* |
| FLOODS | 1608 | 2020 | **418** | **193** *(46%)* |
| ROCKFALLS | 1781 | 2017 | **44** | **37** *(84%)* |
| GLACIAL HAZARD | 1807 | 2020 | **9** | **8** *(89%)* |
| LANDSLIDES | 1941 | 2015 | **6** | **6** *(100%)* |
| CASCADING MULTI-HAZARD | 1807 | 2012 | **12** | **10** *(83%)* |

**Table 6: Summary of the temporal distribution of the events as function of considered hazards, showing for each of them: the total number of events, the oldest and most recent event, the number of events over the period 1920-2020 and the associated percentage. The flood category includes all types of floods (more or less concentrated in sediments) and debris flows.**

Finally, the majority of damaging events were recorded over the period 1920-2020 for the four main categories of damage (Figure 10). Events that caused material damage range from 1628-2020, with higher numbers in the 18th century and from 1920 to 2020. Events that caused functional damage (from 1680 to 2020) follow the same trend, although with fewer events until 1920. With the exception of 3 events (two avalanches on the 27/01/1757 and a rockfall on the 30/05/1856), all the events that caused human damage took place in the 20th and 21st centuries (from 1962 to 2020). Events that caused environmental damage range from 1613 to 2019 and are mainly concentrated in the periods 1674-1807, 1852-1869 and 1926-2019 with a marked decrease since 1970.

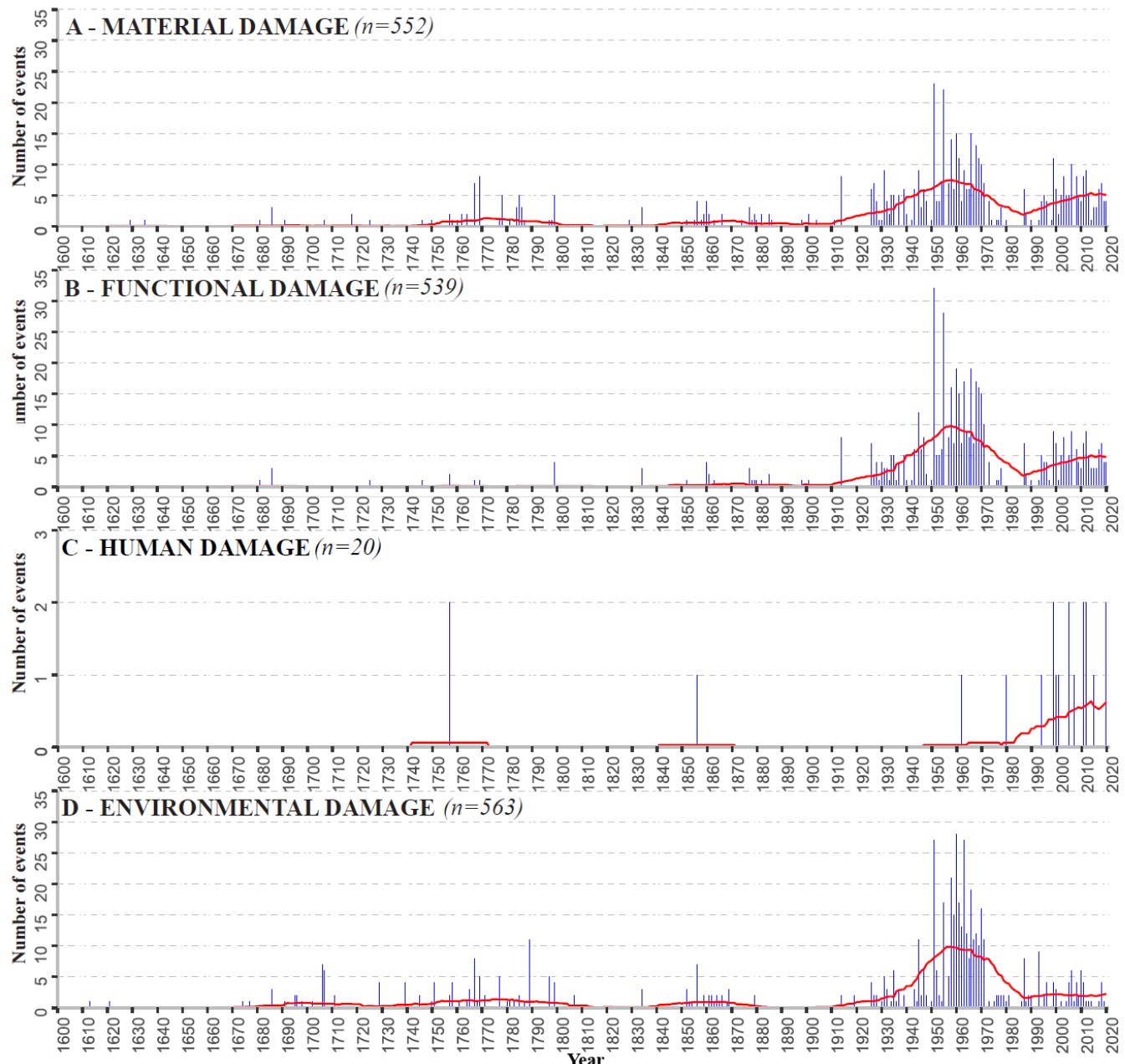

**Figure 10: 1600-2020 chronology of the number of events from the Vallouise-Pelvoux multirisk database as function of damage categories. A- Material damage, B- Functional damage, C- Human damage, D- Environmental damage. The y-axis of the chronology of human damage has been rescaled to make the graph more readable. Each temporal distribution is provided as raw annual counts with a 31-year moving average.**

## 5 Discussion

### 5.1 From composite sources to a multirisk database

This study carried out in the municipality of Vallouise-Pelvoux demonstrates the possibility of developing a comprehensive multirisk database for a given mountain area and over a multi-centennial period from a variety of readily, available and accessible sources, combining scattered information from already existing databases and new specific archival searches. This approach resulted in a compilation both extremely dense, i.e., > 2000 single events on a rather small territory, and over an exceptionally long-time frame, e.g., > 4 centuries. With the aim of having the records as comprehensive as possible, following Giacona et al. (2021), we chose a broad definition of an event, namely not necessarily a disaster, since sources also sometimes report hazards that did not cause damage even if events that caused damage are certainly over-represented in the database, especially in the early period of the record (see Sect. 5.2). By contrast to other attempts which focused on a single

hazard or family of hazards (Stoffel et al., 2005; Wilhelm et al., 2022; Boisson, 2022), the Vallouise-Pelvoux multirisk database provides information on all hydrological and gravitational hazards present in the area and their cascading consequences. In addition, in line with a few recent studies (Zhong et al., 2024; Garcia Hernandez et al., 2017), our results show that the obtained database is as unsegregated as possible considering damage-based and hazard-based approaches. Finally, the methodology proposed is innovative in terms of keeping track of the different sources related to the events, allowing detailed investigations of i) the different, sometimes connected impacts related to the same event and ii) how memory of past events has been retrieved by different data producers.

In details, pre-existing sources have provided a considerable number of events to the database (n=1805, Fig. 3). The 304 events added to the database were floods in all their diversity (n=148) and avalanches (n=137) in majority. These new events are mainly damaging events (81 %), and their distribution is mainly concentrated over the last 100 years (64 %). The contribution of these new events is considerable for floods, accounting for 35 % of all floods in the database and mostly distributed over the period 1600-1919 (55 %). However, the data obtained from archival research was used not only to compile new events, but also to add information to events already recorded in databases and pre-existing documents, and even to correct errors. For example, a lot of transcription mistakes, especially in the EPA and BDRTM, have been corrected thanks to archive documents. Hence, in addition to adding precision to the events, the archival research improved reliability of our database by verifying the events first compiled in pre-existing sources. As a result, more than half (51 %) of the database is made up of events verified in the archives and 15 % are new events resulting from the same searches (Fig. 5) so that in total, 66 % of the database can be considered more reliable.

We chose a primary broad hazard categorisation (avalanches, landslides, etc.) to be able to classify the events from the sources without any subjective choice. These denominations could be found in sources more or less independently of time. The sources often provide a more specific description and typology of hazards, but the latter did actually change over time making it difficult to define homogeneous sub-categories all over the study period. Notably whereas sub-categories are nowadays rather homogenous and normalized between sources, they were much more variable from one source to another earlier on. As an archetypal example, debris flow is a rather « new » concept, at least with regards to the considered > 4 century long study period. Another example is that until the beginning of the 20th century powder snow avalanches where sometimes denoted "dust avalanches". To our knowledge, the seminal reference to « lave torrentielle » (debris flow in French) dates back to Surell (1841). As a consequence, older debris flows could not be described as debris flows by sources. And even for the whole nineteen century, the concept was not disseminated broadly, so that sources rather report mud, gravel, flood deposits, etc. It is only from the early 20th century onward that debris flows were largely recognised as genuine processes and therefore registered as different from other floods. As we did not want to interpret the sources and classify observations as debris flows on the basis of the descriptions at our hand, we did not make a specific debris flow category, but rather merged them with all other floods (with a more or less large contribution of the solid phase). Obviously, when the "debris flow" information was available it has been kept. Therefore, further work targeting the various events of hydrological origin in the database could try to go further in their sub-categorisation, exploiting the information already in the database and crossing it with additional useful data, e.g. typical characteristics of the watersheds such as topography or lithology. Note that the other hazard categories we consider also include subcategories which are not always available in sources: shallow and deep landslides for landslides, powder snow avalanche and dense snow avalanches for avalanches etc., but not always. Again, when available this information was kept.

Regarding impacts, we made specific efforts to document all the different types of impacts the events caused, including physical damage to people, buildings, infrastructures and forests, but also, e.g. accessibility losses (road closures) and disruption of various networks such as electric lines. This all, gives insights in local vulnerability and evolution through time. However, certain types of vulnerabilities such as the broad social and organizational vulnerability of the Vallouise-Pelvoux

community is barely reflected in our damage record that is restricted on what sources say, namely in most of the case damage that are related to in a rather tight and direct way.

The combination of sources made it possible to document glacial hazards and cascading multihazards (Fig. 9), and their consequences, whereas these were not present in pre-existing databases. The majority (8 of the 12 multihazards identified) were indeed linked to glacial or periglacial hazards leading to floods and are therefore mostly recent and related to glacial retreat. Remaining documented cascading events are the result of an avalanche or a debris flow filling a watercourse. Potentially other types of interactions between hazards may have existed, or other pairs of hazards may have existed, but the information was not available in sources, and, again, the database was built with the objective of avoiding any potentially subjective interpretation. Also, within our multirisk approach, interactions between damage, either caused by single or multihazards have been carefully recorded as soon as they were documented by sources, e.g. damage to critical infrastructure that caused various types of economic losses. Documenting such complex interactions between hazards and/or impacts has a clear added value for mountain risk assessment at the territorial scale. For instance, cascading multihazards cause damage which are not necessarily more severe than those from more "classic" hazards. However, as they are often triggered by changing climate conditions and with impact located far away from the event source, they are more unexpected, particularly in the case of floods caused by glacial hazards (and not from meteorological events). For example, among the four cascading floods recorded, three required the evacuation of people (mountaineers, hikers or campers) who had been surprised or trapped by the intensity of the water flow.

## 5.2 Sources, hazard occurrence and other drivers of the event distribution

Only a few events were retrieved before 1700, but we do not assume fewer hazards at that time, but rather an effect of sources. Indeed, as already noted by different authors and despite the efforts of compiling as many different sources as possible, the temporal distribution of events in a database stemming from historical sources remains primarily shaped by the sources used, leading to an increase of the number of events over time (Martin, 1996; Giacona et al. 2019; Sarrazin and Athimon, 2019; Pouzet et al., 2022). Notably, the compilation of pre-existing sources provided events mainly covering the period 1920-2020 (Fig. 6). Only a few of these pre-existing sources cover an earlier period: the RCC, which is the result of research in 1907 and 1913 in the municipal and departmental archives, and the BDRTM, which also includes research in the departmental archives. However, several of the pre-existing sources overlap (Table 4). For example, the BDRTM uses data from Mougin (1931) (LITT.SQ) *La restauration des Alpes*, essentially originating from RCC. Archival research in the municipal and departmental archives and of the ONF-RTM provided data for the entire period studied (Fig. 7). The departmental archives cover the period 1608-1971 and have made it possible to add 233 new events. Above all, these archives have made a major contribution to the verification/correction of events from pre-existing sources (n=968) (Fig. 5). The municipal and ONF-RTM archives mention events over more recent periods: 1879-2015 for the municipality archives of Vallouise-Pelvoux and 1750-2020 for the ONF-RTM archives (Fig. 7). The Vallouise-Pelvoux archives made the greatest contribution in terms of adding new events to the database (Fig. 5). Indeed, these archives had not previously been systematically searched. The events in the ONF-RTM archives are mainly concentrated over the last century of the study period, and represent a significant input to the database over this period.

The content of the database secondly results from the elements at risk and activities, with the distribution and proportion of events over time being closely linked to damage (Fig. 7). In earlier periods, hazards were mainly recorded when they caused damage, whereas today they are recorded more systematically (Martin, 1996; Giacona et al., 2017b). Material, functional and environmental damage are the most numerous in our database and follow the same trends in terms of temporal distribution, partially because the events recorded often caused these different types of damage simultaneously. By contrast, human damage is not very numerous and is mainly recorded over the last century. This effect can be explained by changes in mountain activities and the emergence of mountaineering and other mountain leisure activities (Table 5), as already observed in other

case studies (Boisson et al., 2022; Giacona et al., 2022). The temporal distribution of damage mostly concentrated over the 20th century (Fig. 9) can also be explained by changes in land use, in particular the evolution of agro-pastoral practices, and their subsequent decline in favour of tourism. This trend leads to the increase in settlement size and the development of communication networks, which puts new assets at risk. For some hazards, forest expansion may reduce triggering frequencies and extent (Bourrier et al., 2013; García-Hernández et al., 2017; Zgheib et al., 2020). And, as a result, containing the damage to settlements and infrastructures increase, and potentially increasing damage to forests and related cascading consequences, as documented in our database.

The content of the database and the repartition of the events recorded are obviously also influenced by hazards. The database indeed shows the occurrence of different hazards, and even the presence of cascading multihazards. It should be pointed out first that some temporal distributions are very far from the reality of hazard activity pattern, presumably due to the effect of sources and damage (Laternser and Schneebeli, 2002; Giacona et al., 2022). The over-representation of avalanches since 1919 can be explained in particular by the EPA and the originating archives in the departmental archives, the avalanche identification booklets, systematically surveying given avalanche paths. There are fewer rockfalls events (Table 6), but they are spread over the longest period after floods and avalanches. These hazards have often only been recorded if they caused damage, which suggests that there may have been many more events in the past than are represented in the database. Floods are the most numerous and oldest events, with a distribution that shows a more limited increase over time. The explanation is that this hazard has the greatest impact on the valley activities, leading to a large production of archives, notably in the registers of municipal deliberations (Fig. 3). Hence, the chronology of floods may be closer to process activity. Indeed, beyond the effect of sources and damage, there are also physical effects linked to the climate, lithology and topography of the study area that explain event records. For example, the physical characteristics of the commune, notably its cryosphere, explain the presence of avalanches, glacial and periglacial hazards, and have an impact on river regimes and therefore, e.g., the seasonality of floods. Recent climate change may therefore have affected local process activity in the case study, as in close locations (e.g., Eckert et al., 2013; Mainieri et al., 2023), and this influence may result in some characteristics of event chronologies. However, the effect of sources is so pronounced that contextualization and bias-correction techniques need to be applied first to make such changes - if any - detectable (Giacona et al., 2021; 2022).

Regarding multihazards, "only" six pairs could be documented, which may be seen as low. Yet, potentially, other pairs of hazards could have been linked but their identification would require additional data and work. However, it is even not obvious that the actual proportion of multihazards in the record is really low, because, to our knowledge, reference rates do not really exist in the literature, at least for similar case studies, and it is not necessarily dubious that the number of single avalanches or floods is much higher. However, as the record is biased towards damaging events that directly impacted the society, especially far back in the past, it is presumable that some "parts" of cascading hazards that have occurred far away from elements at risk at old times were not reported by the sources, e.g. some of the early floods may have been caused by glacial hazards but we simply do not have the information. This could suggest that the proportion of cascading hazards may truly be a bit underestimated. To ascertain this assumption, it could be possible to investigate in more details events for which cascading hazards may have occurred, looking for evidence in extra-data, e.g., to search for narrative testimonies of photographs of glaciers surface that may suggest the rupture of an underlying water pocket at a time for which a flood exist in our record that does not correspond to intense precipitation or another evident meteorological driver. In parallel a more in-depth study of the perception of hazards by ancient local societies could be conducted, to try to grasp evidences that they perceived or not potential linkage between hazards.

The same more or less applies to cascading impacts, e.g. it is likely that their proportion (and arguably diversity) is underestimated in the database as the whole damage information could certainly not be extracted from the sources, especially for the old times. An interesting further work would therefore be to i) try to expand the documentation of cascading impacts from additional resources, expand the analysis of the most complex events of the database, ii) try to get the picture as complete

as possible of multirisk patterns and their change in the study area, taking into account changes in the ways interactions between hazards and/or damage were perceived or not.

All in all, the shape of event chronologies results in a complex combination of sources, exposure, wider social practices and perceptions and process activity, so that understanding the shape in the chronologies involves an in-depth analysis to disentangle the different effects. At this stage, all specific patterns of the different chronologies cannot therefore be fully explained. For example, the 1950-1970 peak in the annual number of events may be related to a combination of records more comprehensive than for earlier periods and numerous new elements at risk related to tourism development, whereas the recent decrease could be linked to better protection measures for these new elements at risk combined to anthropogenic warming (Jacquemart et al., 2024). Further in-depth joint analyses of events and various context data could ascertain or contradict these intuitions.

Finally, among the main features of the spatio-temporal distribution of the events and their underlying causes, some elements may be specific to the study area, whereas some other may be of broader relevance. Regarding mobilized sources, we know that all pre-existing sources can be used in other French Alpine territories (BDMVT, BDC2ROP, PPR, LITTERATURE, CD05). BDRTM, EPA and CLPA can also be used in the Pyrenees. As literature sources are more specific and local, they will be all the more variable from one place to another. Concerning the input of archival searches, Vallouise-Pelvoux has numerous and well conserved documents, which allowed a great contribution to the database and has provided a continuous source of information over the entire time period. However, archive conservation in each municipality or department, and the absence of destruction and losses due, e.g., to disasters, fires or wars is largely variable from one location to another (Sanchez-Garcia, 2023). As a matter of fact, the sources used here to constitute the database can be quite reused on other French alpine municipalities, whereas the overall methodology of merging pre-existing databases with more specific archival searches to generate a multi risk database as comprehensive as possible can be transferred in more or less all mountain territories with ancient human occupation. Beyond sources, results in terms of spatio-temporal distribution of hazards, events, and damage typology depend on topography, prevailing climate and land use conditions, human activities and their changes over the considered period. Hence, some patterns may truly be rather generic over large areas, such as those related to the transition from an agro-pastoral society to a society dominated by tourism and to the shrinkage of the cryosphere. However, local factors may also play a strong role, so that detailed findings in terms of, e.g., prevalent hazard and damage type, their causes and their pattern of change may be quite different from one context to another, even at close locations (Zgheib et al., 2020).

## 6 Conclusions and wider outlooks

The current context of exacerbated climate and socio-environmental changes calls for mountain risk management strategies holistic enough to allow anticipation. A prerequisite for their design is i) the documentation as exhaustive as possible of mountain risks at the territorial scale (Jacquemart et al., 2024) and ii) the analysis and exploitation of this information to decipher the complex dynamics at play. Our results obtained in the example of Vallouise-Pelvoux, an archetypal municipality of the Alpine space, suggest that an effort to combine existing records with archival searches within a unique multirisk database may be a step into the right direction. Notably, the multirisk nature of the approach makes it possible to address the risk system in its entirety, namely to consider i) both hazards and their various impacts, and ii) potential interactions between hazards and/or damage. It would therefore be useful/possible to transfer the approach to other Alpine valleys. This would, among other findings, provide a better understanding of what can be generalised to other territories and what is specific to the studied context (availability of data sources as function of time, hazard and damage type, respective weight of the different hazard and damage, *etc.*).

Despite the exceptional richness of the Vallouise-Pelvoux Database, there is certainly room for improvements to reach a more comprehensive documentation of past events, both in terms of event numbers and their characteristics. For example, research at the departmental archives has focused mainly on the period 1600-1950, but there are still additional documents for the period 1950-2020 that could be exploited. Another way to complete the data would be to include paleoenvironmental data related to past events (e.g., from tree rings, or sedimentological cross-sections in alluvial fan studies, Stoffel and Bollschweiler, 2008; Corona et al., 2010; Schläppy et al., 2013; Wilhelm et al., 2022). However, even if our results already show that the spatiotemporal distribution of events retrieved from a combination of sources over a long period is shaped by a complex combination of sources and other social and physical factors, the priority is probably to expand the analysis to reveal the added value of the data to assess local changes in risks, or to identify hazard-climate trends. First hints for further investigations have already been provided but in one word, main outlook is to complement the event database with refined biophysical and socio-economic context data (Martin et al., 2015), to produce a comprehensive geohistorical analysis of the factors at play (Giacona et al., 2019b; 2022) and to potentially use bias correction techniques to move from events to hazards and risks (Giacona et al., 2021). At this cost, valuable input regarding long-term changes in mountain risk systems and their components may be delivered, ultimately contributing to the sustainability of mountain communities.

*Data availability*: The data underlying this research is freely available at: Dallons Thanneur, L., Florie, G., Frey, P., & Eckert, N. (2025). Constitution of a multicentennial multirisk database in a mountainous environment from composite sources: the example of the Vallouise-Pelvoux municipality (Ecrins, France) - source data [Data set]. Zenodo. *https://doi.org/10.5281/zenodo.14848820*

*Competing interests:* The authors declare that they have no conflict of interest.

*Author contribution:* FG, NE and PF designed the research. LDT and FG designed the archival searches. LDT conducted the archival searches, produced the statistics and illustrations. All co-authors wrote and edited the manuscript.

*Financial support:* L. Dallons Thanneur was supported by a PhD grant from INRAE and Labex OSUG. Further financial support came from the French National Research Agency to the IRIMONT program (**ANR-22-EXIR-0003**). F. Giacona, N. Eckert and P. Frey are members of the Grenoble Risk Institute (https://risk.univ-grenoble-alpes.fr/fr ). IGE is member of Labex OSUG and Labex Tec21.

*Acknowledgments:* Archival research was carried out at the Hautes-Alpes departmental archives (Gap, France), the Vallouise-Pelvoux municipal archives (Vallouise-Pelvoux, France) and the ONF-RTM archives (Gap, France). Authors are grateful to the numerous people that contributed to build, preserve and give access to the archives, and to J. Gill and an anonymous referee whose insightful comments helped producing a better paper.

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
