# Peer review of "Constitution of a multicentennial multirisk database in a mountainous environment from composite sources: the example of the Vallouise-Pelvoux municipality (Ecrins, France)"

_EGUsphere, 2025_

## Referee Comment (RC1)

**Reviewer Comments**

**Constitution of a multicentennial multirisk database in a mountainous environment from composite sources: the example of the Vallouise-Pelvoux municipality (Ecrins, France)**

https://doi.org/10.5194/egusphere-2025-761

**Summary (General Comments):** This is a very interesting manuscript, with a clear aim and methodology. The latter is well executed, with a comprehensive set of results and evidenced discussion. The authors make very good use of existing literature to support most of their writing. Overall, the manuscript offers an interesting and useful addition to the literature. It demonstrates well the potential of blending multiple evidence sources to develop richer understanding of hazard events. Below I note some specific and technical comments that I believe would strengthen this manuscript.

**Specific Comments:**

**1. Multirisk.** You have developed an important and exciting database, that contains information about multiple hazards and cascading multi-hazards, and their impacts. However, some reflection is needed on the framing of this as a 'multirisk' database. There is no universally accepted definition of the term 'multirisk' (though multi-hazard does have a UNDRR included definition). Previous work has explored what others have meant by this term, and generally accepted that multirisk includes reflection on interrelationships at the vulnerability as well as hazard level (https://www.myriadproject.eu/wp-content/uploads/2022/11/D1_2_Handbook.pdf). The reference to Curt (2021) on Line 39 is perhaps rather limited given this was a bibliometric analysis. If you are going to pitch this as a 'multirisk' database then I think there needs to be greater engagement with a broader body of multirisk literature throughout this manuscript, explaining what is understood by this approach and how this work aligns with and builds on existing work.

**2. Clarity regarding cascading multi-hazards.**

> **a. Literature.** The manuscript would benefit from a greater engagement with the multi-hazard and multi-risk literature and inclusion of a definition of 'multi-hazard' (https://www.undrr.org/terminology/hazard#:~:text=Multi%2Dhazard%20means%20(1),account%20the%20potential%20interrelated%20effects) in the opening sections.

> **b. Database.** The manuscript would benefit from clarity as to how cascading multi-hazards are being identified and included in the database. Section 3.4.1, for example, is quite vague – noting that these events are distinct in the database but without a full and detailed explanation as to why and how these are identified. In the results, **Table 6** is the only time 'cascading multi-hazard' is included as its own row. It's unclear if this is merged into other categories elsewhere – or omitted.

> **c. Discussion.** There is limited critical reflection in the discussion section about the cascading multi-hazards, including (for example) how few of them there are. Are there genuinely very few cascading multi-hazard events in this region – or is this a function of not having the information to categorise events as being 'cascading multi-hazard', a function of your definition of 'cascading multi-hazard'? I was hoping for more reflection on this in the discussion, and suggestions of how you could explore this further.

**3. Vulnerability.** There is only one reference to vulnerability in the entire manuscript (Line 284). Given the prominence of vulnerability to risk, and interactions at the vulnerability level to many definitions of

multirisk this is surprising. There is scope for greater reflection on the concept of vulnerability (https://www.undrr.org/terminology/vulnerability) throughout this manuscript – how it may have changed over time, how it may also be responsible for changes in disaster impacts (alongside changes in exposure, for example).

**Technical Comments:**

**Line 90-93.** Include specific sections (Sect 2. Etc) to improve signposting in the manuscript.

**Figure 1.** You note where these images are from, but not the license by which you are reproducing them / confirmation that you have permission to reuse them.

**Line 120.** A source is needed for the information about the surface of glaciers in 2018.

**Line 195-199.** You make reference to the C Series, O Series etc – but it's not clear here what the 'C' or 'O' relates to. An additional sentence or two here to clarify would be useful. Alternatively, signpost to a later section where more information is given.

**Figure 2.** The figure caption would benefit from a little more detail about the relevance of the alphanumerical codes (e.g., 2C, 1M). The significance of the red writing (difficulties in readability...) is also unclear.

**Line 275.** The meaning of 'mobilized sources' is not clear.

**Line 279.** In addition to saying 'oldest periods' can you give a date range, to support the reader.

**Line 325.** Given 'risk' has quite a specific meaning, should the wording "range of information on risks related to various hazards" actually be something such as: "range of information to characterise various hazards and their impacts".

**Line 360.** Section 4 would benefit from a short opening paragraph outlining what each sub-section will present.

**Line 368.** 'The mentions' is not clear given the preceding paragraph. Can you be more specific (all mentions or a sub-set of mentions etc).

**Line 390.** This is the first point I realise that debris flows are included as part of your 'floods' category. I think this needs to be made clearer earlier on, and justified. Why are they included within this category and not as their own stand-alone category? Are there other hazard 'sub-types' included in each of the hazard categories that you have used? If so – can you explain these earlier in the writing.

**Line 392.** I don't think 'small half' is suitable language. Consider 'just under half' instead.

**Line 428.** Change synthetizes to synthesizes

**Line 431.** I don't think it is possible to use the data in Table 4 to say *'caused the most damage*' rather '*the most events causing some level of damage*' (or equivalent wording) – your numbers don't take into account whether that damage was small or large.

**Line 450.** There is a word missing in the phrase '1,674 damage recorded'. Should it be damage types, or damage categories?

**Table 5.** Check the spelling of words in this table. Materiel vs. material; vehicules vs vehicles; stuctures vs structures.

**Line 466.** The phasing 'during which mentions numerous events' is unclear.

**Line 544** (and throughout). Check if a space is needed between a number and % sign.

**Line 589.** Should 20e century be 20th century?

**Line 620**. Suggest avoid using the term 'natural disasters' given current dialogue about it not being an appropriate term. The word 'disasters' should be sufficient.

---

## Author Comment (AC1)

**Referee 1**

**1) General comments**

This is a very interesting manuscript, with a clear aim and methodology. The latter is well executed, with a comprehensive set of results and evidenced discussion. The authors make very good use of existing literature to support most of their writing.

Overall, the manuscript offers an interesting and useful addition to the literature. It demonstrates well the potential of blending multiple evidence sources to develop richer understanding of hazard events.

Below I note some specific and technical comments that I believe would strengthen this manuscript.

Authors' response: We deeply thank Joel Gill for his encouragements and meaningful suggestions that will greatly help us to improve the paper. In what follows, we provide a point-by-point answer to his comments, questions and suggestions.

**2) Clarity regarding cascading multi-hazards**

**a. Literature.**

The manuscript would benefit from a greater engagement with the multi-hazard and multi-risk literature and inclusion of a definition of 'multi-hazard' (https://www.undrr.org/terminology/hazard#:~:text=Multi%2Dhazard%20means%20(1 ),account%20the%20potential%20interrelated%20effects) in the opening sections.

Authors' response: A multitude of terms has been proposed to account for increasingly complex hazards and risks, e.g. multi-hazards, cascading hazards, domino effects, compound events, cascading risks, interconnected risks, etc. (e.g. Simpson *et al.*, 2021; Zscheischler *et al.*, 2018). Despite the slight difference in meaning between these different terms, they all emphasize the importance of interactions between hazards and/or exposed elements. In the paper, we use the multirisk concept in a broad sense, namely all single risks and their interactions at the territorial scale, with interactions concerning potentially hazards and/or impacts. Similarly, we define multi-hazards as all single hazards and their interactions at the territorial scale. These definitions are indeed consistent with the one of UNDRR (2017). In the reworked version of the paper, we will make an extra-effort to clarify these definitions and make reference to the one of UNDRR.

**b. Database.**

The manuscript would benefit from clarity as to how cascading multi-hazards are being identified and included in the database. Section 3.4.1, for example, is quite vague – noting that these events are distinct in the database but without a full and detailed explanation as to why and how these are identified. In the results, Table 6 is the only time 'cascading multi-hazard' is included as its own row. It's unclear if this is merged into other categories elsewhere – or omitted.

Authors' response: First, let us recall that the database was built with the objective to avoid any subjectivity, restricting strictly the information recorded to what the sources actually say. Regarding multi-hazards (see our response above regarding definitions), interactions that sources identified correspond all to cascading hazards, namely the 12 pairs of successive hazards for which causal linkages, where reported by the sources. Potentially other types of interactions between hazards may have existed, or other pairs of hazards may have succeed, but the information was not available. In the reworked version of the paper, we will define more clearly cascading hazards as a specific type of multi-hazards, and the sole that could be documented from the analyzed sources.

**c. Discussion.**

There is limited critical reflection in the discussion section about the cascading multi-hazards, including (for example) how few of them there are. Are there genuinely very few cascading multi-hazard events in this region – or is this a function of not having the information to categorize events as being 'cascading multi-hazard', a function of your

definition of 'cascading multi-hazard'? I was hoping for more reflection on this in the discussion, and suggestions of how you could explore this further

Authors' response: See our previous response about subjectivity and the type of interactions in hazards that the sources documented. Potentially, other pairs of hazards could have been linked (or not) but it is impossible to know it with certainty without additional data/information/work. So we cannot fully explain the relatively low number of cascading events in the record so far. However, it is even not obvious that the actual proportion of cascading hazards in the record is really low, because, to our knowledge, reference rates do not really exist in the literature, at least for similar case studies, and it is not necessarily dubious that the number of single avalanches or floods is much higher. Yet, as the record is biased towards damaging events that directly impacted the society, especially far back in the past, it is presumable than some "parts" of cascading hazards that have occurred far away from elements at risk at old times were not reported by the sources, e.g. some of the early floods may have been caused by glacial hazards but we simply do not have the information. This could suggest that the proportion of cascading hazards may be a bit underestimated. To go further, it could be possible to investigate in more details events for which cascading hazards may have occurred, looking for evidence in extra-data. E.g., search for narrative testimonies of photographs of glaciers surface that may suggest the rupture of an underlying water pocket at a time for which a flood exist in our record that does not correspond to intense precipitation or another evident meteorological driver. In parallel a more in-depth study of the perception of hazards by ancient local societies could be conducted, to try to grasp evidences that they perceived or not or not potential linkage between hazards. The discussion of the reworked paper will include these elements.

**3) Technical Comments**

Line 90-93. Include specific sections (Sect 2. Etc) to improve signposting in the manuscript.

Authors' response: The resided paper will be reworked as follows: "In the following, the method used to produce the Vallouise-Pelvoux multirisk database is described (Sect. 3). The dataset and a statistical overview of sources, events and their characteristics are then presented (Sect. 4). Finally, the main characteristics of the spatio-temporal distribution of the events and their drivers (Sect. 5) and potential outlooks (Sect. 6) are discussed".

Figure 1. You note where these images are from, but not the license by which you are reproducing them / confirmation that you have permission to reuse them.

Authors' response: The photographs used come from public archives and can therefore be used freely, especially for research purpose, without the need of an additional license.

Line 120. A source is needed for the information about the surface of glaciers in 2018.

Authors' response: This number was obtained from a careful processing of existing aerial photographs and remote sensing images (Defernand, 2021). We will add the reference and inform about the method used in the revised version of the paper.

Line 195-199. You make reference to the C Series, O Series etc – but it's not clear here what the 'C' or 'O' relates to. An additional sentence or two here to clarify would be useful. Alternatively, signpost to a later section where more information is given.

Authors' response: These letters (sometimes associated to numbers) correspond to the coding system used by the French archives and they do not have an extra specific meaning (e.g. a letter is not the capital letter of a given word). In the revised version of the paper, we will add the precision, and also, at this point of the paper, indicate that the different archive series are detailed in Sect. 3.3.

Figure 2. The figure caption would benefit from a little more detail about the relevance of the alphanumerical codes (e.g., 2C, 1M). The significance of the red writing (difficulties in readability…) is also unclear.

Authors' response: Regarding the archive series, we will add in the figure caption the same precision as in text (see our previous response). Regarding the red boxes (drawn by us), we will indicate "red boxes were drawn to highlight the segments of original text for which a translation is provided".

Line 275. The meaning of 'mobilized sources' is not clear.
Authors' response: We will add the reference to the section where it is defined in the revised version of the paper: « This first set of pre-existing sources used (sect. 3.2), although substantial, presents weaknesses."

Line 279. In addition to saying 'oldest periods' can you give a date range, to support the reader.
Authors' response: We will add the precision in the revised version of the paper: "we notice a lack of documentation for the oldest periods (1600-1900) which are very important to analyze hazard evolution through time and space."

Line 325. Given 'risk' has quite a specific meaning, should the wording "range of information on risks related to various hazards" actually be something such as: "range of information to characterize various hazards and their impacts".
Authors' response: We will make the correction in the revised version of the paper: "The multirisk database gathers a range of information to characterize various hazards and their impacts: hydrological hazards, rockfalls, avalanches, landslides and glacial hazards". See also our response to the second comment of anonymous referee 2 about risk.

Line 360. Section 4 would benefit from a short opening paragraph outlining what each sub-section will present.
Authors' response: We will add the following opening paragraph in the revised version of the paper: "In Sect. 4.1, the 3,528 hazard mentions gathered from the various sources investigated are presented. Since different mentions may refer to the same hazard and impacts, the whole corpus results in 2,131 single events introduced in Sect. 4.2. Sect 4.3 describes the overall characteristics of these events, while Sect.4.4. focuses on their various impacts and Sect. 4.5 on their temporal distribution."

Line 368. 'The mentions' is not clear given the preceding paragraph. Can you be more specific (all mentions or a sub-set of mentions etc).
Authors' response: We will rework the sentence in the revised version of the paper: The 3,528 mentions concern mostly avalanches (n=2654), floods (n=762) and rockfall (n=88). By contrast, landslides (n=15) and glacial hazards (n=9) represent lower proportions of mentions (Fig. 4B).

Line 390. This is the first point I realise that debris flows are included as part of your 'floods' category. I think this needs to be made clearer earlier on, and justified. Why are they included within this category and not as their own stand-alone category? Are there other hazard 'sub-types' included in each of the hazard categories that you have used? If so – can you explain these earlier in the writing.
Authors' response: Debris flow is a rather « new » concept, at least with regards to the considered > 4 century long study period. To the authors' knowledge, the seminal reference to « lave torrentielle » (debris flow in French) dates back to Surell (1841). As a consequence, older debris flows could not be described as debris flows by sources. And even for the whole nineteen century, the concept was not disseminated broadly, so that sources rather report mud, gravels, flood deposits, etc. It is only from the early 20th century onward that debris flows were largely recognised as genuine processes and therefore registered as different from other torrential floods. At this stage of the work, we did not want to interpret the sources and classify observations as debris flows on the basis of the descriptions at our hand, which would have been at least partially subjective. As a

consequence, for the consistency of the record over the whole study period, we did not make a specific debris flow category, but rather merged them with all other events of hydrological origin as floods, with a more or less large contribution of the solid phasis. Obviously, when the "debris flow information" was available it has been kept. Therefore, further work targeting the various events of hydrological origin in the database could try to go further in their sub-categorisation, exploiting the information already in the database and crossing it with additional useful data, e.g. typical characteristics of the watersheds such as topography or lithology. Note that the other hazard categories we consider also include subcategories which are not always available from sources: shallow and deep landslides for landslides, powder snow avalanche and dense snow avalanches for avalanches etc., but not always. Again, when available this information was kept.

In the revised version of the paper, we will precise early on in the paper that debris flows are included within the flood category and better justify why we primarily chose a broad hazard categorisation. We will also include within the discussion a point about the report of hazard types by sources including the emergence of the debris flow concept during the nineteen century.

Line 392. I don't think 'small half' is suitable language. Consider 'just under half' instead.
Authors' response:  We will write 'just under half' in the revised version of the paper.

Line 428. Change synthetizes to synthesizes
Authors' response: We will make the correction in the revised version of the paper.

Line 431. I don't think it is possible to use the data in Table 4 to say 'caused the most damage' rather 'the most events causing some level of damage' (or equivalent wording) – your numbers don't take into account whether that damage was small or large.
Authors' response: We will make the correction in the revised version of the paper: « [...] however, the most events causing some level of damage are floods, followed by rockfall, landslides and avalanches."

Line 450. There is a word missing in the phrase '1,674 damage recorded'. Should it be damage types, or damage categories?
Authors' response: We will make the correction in the revised version of the paper: "Of the 1,069 events that caused damage, there was a total of 1,674 damage recorded in the four major categories, [...]"

Table 5. Check the spelling of words in this table. Materiel vs. material; vehicules vs vehicles; stuctures vs structures.
Authors' response:  We will make all the required corrections in the revised version of the paper.

Line 466. The phasing 'during which mentions numerous events' is unclear.
Authors' response: We will rewrite the sentence  in the revised version of the paper as « We can also note other rich periods, 1608-1640, 1750-1800 and 1850-1870, although less rich than the 1920-2020 period"

Line 544 (and throughout). Check if a space is needed between a number and % sign.
Authors' response: yes, we will make the correction in the revised version of the paper.

Line 589. Should 20e century be 20th century?
Authors' response: We will make the correction in the revised version of the paper: "The temporal distribution of damage mostly concentrated over the 20th century [...]"

Line 620. Suggest avoid using the term 'natural disasters' given current dialogue about it not being an appropriate term. The word 'disasters' should be sufficient.

Authors' response: We will make the correction in the revised version of the paper:"[...] and the absence of destruction and losses due to disasters [...]"

**References:**

Defernand, C. (2021). Analyse diachronique de l'occupation des sols par classification d'images multi-sources d'une haute vallée alpine du massif des Ecrins. Master thesis. Master de Sciences Humaines et Sociales. Parcours GÉOgraphie, Information, Interface, Durabilité, EnvironnementS. Université Grenoble Alpes, 50p.

Simpson, N. P., Mach, K. J., Constable, A., Hess, J., Hogarth, R., Howden, M., ... & Trisos, C. H. (2021). A framework for complex climate change risk assessment. *One Earth*, *4*(4), 489-501.

Surell, A. (1841). Etude sur les torrents des Hautes Alpes, Paris, France, 280p.

United Nations Office for Disaster Risk Reduction - UNDRR (2017). The Sendai Framework Terminology on Disaster Risk Reduction. "Hazard". Accessed 28 June 2025. https://www.undrr.org/terminology/hazard.

Zscheischler, J., Westra, S., Van Den Hurk, B. J., Seneviratne, S. I., Ward, P. J., Pitman, A., ... & Zhang, X. (2018). Future climate risk from compound events. *Nature climate change*, *8*(6), 469-477.

---

## Author Comment (AC2)

**Referee 2**

**1) General comments (Anonymous Referee #2)**

I would like to thank the authors for a highly interesting and original manuscript. I very much enjoyed reading it.

Authors' response: We deeply thank the referee for his/her encouragements and meaningful suggestions that will greatly help us to improve the paper. In what follows, we provide a point-by-point answer to his/her comments, questions and suggestions.

**2) Comments**

Page 2: "A specific concern is related to complex and/or cascading hazards that can have far-reaching and multiple consequences downstream" -> I think it would be good to provide a very clear definition of multi, consecutive, cascading, etc risk. Similarly, a couple of sentences down (risks related to natural hazards are broadly defined here as the results of interactions between natural and societal components), I wonder if it would be good to explicitly define is as exposure and vulnerability (and risk) very clearly? And eg in L. 329 you talk about "cascading multi-hazards".

Authors' response: A multitude of terms has been proposed to account for increasingly complex hazards and risks, e.g. multi-hazards, cascading hazards, domino effects, compound events, cascading risks, interconnected risks, etc. (e.g. Simpson *et al.*, 2021; Zscheischler *et al.*, 2018). Despite the slight difference in meaning between these different terms, they all emphasize the importance of interactions between hazards and/or exposed elements. In the paper, we use the multirisk concept in a broad sense, namely all single risks and their interactions at the territorial scale, with interactions concerning potentially hazards and/or impacts. Similarly, we define multi-hazards as all single hazards and their interactions at the territorial scale. Interactions that sources identified correspond all to cascading hazards, namely the 12 pairs of successive hazards for which causal linkages where reported. Potentially other types of interactions between hazards may have existed, but the information was not available. In the reworked version of the paper, we will make an extra-effort to clarify these definitions.

Regarding our risk definition, it was similarly chosen to be broad enough to easily account for elements that are not directly explicit in the classical IPCC risk decomposition between hazard, vulnerability and exposure, such as spatio-temporal dynamics, delayed interactions etc. Again, the reworked version of the paper will make this even more explicit.

The authors speak of a multirisk database that covers centuries, but it covers hazards and impact, right? It would be good to clarify this; I also found section 3.4.1 a bit unclear.

Authors' response: Our formulation was indeed a bit awkward. In the revised version of the paper we will reformulate the sentence as: "The multirisk database gathers a range of information to characterize various hazards and their impacts over a pluricentennial time frame: hydrological hazards, rockfalls, avalanches, landslides and glacial hazards". We will also rework the whole paragraph to make it clearer.

Out of curiosity: have definitions of individual hazard types changed over time (eg land movement or rockfall)?

Authors' response: We chose a primary broad hazard categorisation (avalanches, landslides, etc.) to be able to classify the events from the sources without any subjective choice. These denominations could be found in sources more or less independently of time. However, the sources often provide a more specific description and typology of hazards that did actually change over time and makes it difficult to define homogeneous sub-categories all over the study period. Notably whereas sub-categories are nowadays rather homogenous and normalized between sources, they were much more variable from source to another earlier on. See also our response to referee 1 about the emergence of the debris flow concept during the nineteen century. Another example is that until the beginning of the 20th century powder snow avalanches where sometimes denoted "dust avalanches". In the revised version of the paper we will better justify our broad hazard categorization and discuss the changes in sub-typologies provided by sources as function of time.

**4.2: by event, do you mean disaster? (so when a hazard interacts with exposure and vulnerability of that exposure)? In the next section you talk about impacted assets, but does this mean that you exclude vulnerability?**

Authors' response: As stated in the paper, following Giacona et al. (2021) 'event' means the occurrence of a given hazard and its characteristics, including impacts retrieved from sources. This broad definition means that an event is not necessarily a disaster, since sources also sometimes report hazards that did not cause damage. With the aim of having the records as comprehensive as possible, we also keep such events. However, it is true that events that caused damage are over-represented in the database, especially in the early period of the record. This is common in hazard records resulting from historical sources as disasters are generally more systematically recorded and better preserved by sources than events that did not impact societies (Giacona et al., 2017).

Regarding impacts, we made specific efforts to document all the different types of impacts the events caused, including physical damage to people, buildings, infrastructures and forests, but also, e.g. accessibility losses (road closures) and disruption of various networks such as electric lines. This all in turn informs regarding local vulnerability and how it evolved through time (See also our response about the hazard/vulnerability/exposition of risk). However, certain types of vulnerabilities such as the broad social and organizational vulnerability of the Vallouise-Pelvoux community is barely reflected in our damage record. More broadly, our work is not a comprehensive study of the local vulnerability (as it is not a comprehensive study of the local risk system or of local hazard activity). This could be the goal of further work, for which the information summed up will be a good starting point, to be combined with other information. In the revised version of the paper we will expand the outlook section to include this perspective.

**Looking at the analysis in section 4.5, I wondered why we see such a peak in number of disasters around 1950-1970?**

Authors' response: As stated in the paper the database context and especially the shape of event chronologies results in a complex combination of sources, exposure and wider social practices and process activity. Understanding the shape in the chronologies involves an in-depth analysis to distangle the different effects, which is beyond the analysis carried so far (but is a priority for future work). At this stage we therefore do not have a clear explanation for this 1950-1970 peak, except that arguably since the mid 20th century we probably have a much more comprehensive record as for earlier periods, which combines with numerous new elements at risk related to tourism development. The recent decrease could be linked to better protection measures for this new elements at risk combined to anthropogenic warming. But again this explanation is rather speculative at this stage. IN the revised version of the paper, we will indicate that the 1950-1970 peak is an excellent example of a pattern that an in-depth analysis of event and context data could explain, to determine if it is related to changes in sources, exposure and/or process activity.

**I would be curious to read a bit more of an in depth/critical reflection on and discussion of the notion of single vs (cascading?) multi-risk and the recording (and hence perception) thereof.**

Authors' response: First, as already stated the database was built with the objective to avoid any subjectivity, restricting strictly the information recorded to what the sources actually say. Regarding multi-hazards, this resulted in 12 cascading hazards "only" (see our response above regarding definitions). Potentially other types of interactions between hazards may have existed, or other pairs of hazards may have succeed, but the information was not available. Second, within our multirisk approach, interactions between damage are also considered, and as soon as they have been mentioned in the sources they have been carefully kept, e.g. damage to critical infrastructure that caused various types of economic losses. These cascading consequences are not analyzed in the paper which is already long. See our response to Joel Guil about perception of multi-hazards. The same more or less applies to cascading impacts, e.g. it is likely that their proportion (and arguably diversity) is underestimated in the database as the whole information could

certainly not be extracted from the sources, especially for the old times. An interesting further work would therefore be to i) try to expand the documentation of cascading hazards/impacts from additional resources, expand the analysis of the most complex events of the database (e.g. those with cascading impacts), iii) try to get the picture as complete as possible of multirisk patterns and their change in the study area, taking into account changes in the ways interactions between hazards and/or damage were perceived or not. Of course this is something very ambitious that will require lots of additional research. In the reworked version of the paper, we will include some discussion elements on this purpose.

**3) Minor comments**
7 L. 190: studied period -> should be study period
Authors' response: We will make the correction in the revised version of the paper: "The variety of pre-existing and new sources mobilized in the constitution of the database allowed us to cover the 420-year study period"

275: "mobilized sources" -> do you mean used sources?
Authors' response: We will make the correction in the revised version of the paper: "This first set of pre-existing sources used (sect. 3.2), although substantial, presents weaknesses."

293: "torrents" -> do the authors mean torrential rainfall?
Authors' response: no it means torrents as river, generally of moderate size, and with a slope greater than 6%, with specific flood regimes (often highly charged in sediment". We will add a note in the revised version of the manuscript.

You write multirisk and multi-hazard, I would hyphenate both.
Authors' response: Right. For consistency we will use "multirisk" and "multihazard" in the revised version of the paper, with clear definitions at the first occurrence of the term.

636: I am not sure I would call the frequency figures a statistical analysis.
Authors' response: True, our formulation was a bit "too much" also it was referring not only to the chronologies of events. We will reformulate as: "Indeed, the first analyses (distribution of events according to sources, space, time, damage etc.) already highlight the potential of the information for more in-depth studies.

**References:**
Giacona, F., Eckert, N., & Martin, B. (2017). A 240-year history of avalanche risk in the Vosges Mountains based on non-conventional (re) sources. *Natural Hazards and Earth System Sciences*, *17*(6), 887-904.

Giacona, F., Eckert, N., & Martin, B. (2022). Comment interpréter une chronologie événementielle en géohistoire? L'exemple de deux siècles et demi d'avalanches dans le Massif vosgien. *Cybergeo: European Journal of Geography*.

Simpson, N. P., Mach, K. J., Constable, A., Hess, J., Hogarth, R., Howden, M., ... & Trisos, C. H. (2021). A framework for complex climate change risk assessment. *One Earth*, *4*(4), 489-501.

Zscheischler, J., Westra, S., Van Den Hurk, B. J., Seneviratne, S. I., Ward, P. J., Pitman, A., ... & Zhang, X. (2018). Future climate risk from compound events. *Nature climate change*, *8*(6), 469-477.